# TTT-UNet: Enhancing U-Net with Test-Time Training Layers for Biomedical Image Segmentation

**Rong Zhou**[*][1]                                        roz322@lehigh.edu
**Zhengqing Yuan**[*][2]                                 zyuan2@nd.edu
**Zhiling Yan**[*][1]                                     zhy423@lehigh.edu
**Weixiang Sun**[*][2]                                    wsun4@nd.edu
**Kai Zhang**[1]                                          kaz321@lehigh.edu
**Yiwei Li**[3]                                           yl80817@uga.edu
**Yanfang Ye**[2]                                         yye7@nd.edu
**Xiang Li**[4]                                           xli60@mgh.harvard.edu
**Lichao Sun**[1]                                         lis221@lehigh.edu
**Lifang He**[1]                                          lih319@lehigh.edu

[1] *Lehigh University, Bethlehem, PA, USA*

[2] *University of Notre Dame, Notre Dame, IN, USA*

[3] *University of Georgia, Athens, GA, USA*

[4] *Massachusetts General Hospital and Harvard Medical School, Boston, MA, USA*

**Editors:** Accepted for publication at MIDL 2026

## Abstract

Biomedical image segmentation is crucial for accurately diagnosing and analyzing various diseases. However, Convolutional Neural Networks (CNNs) and Transformers, the most commonly used architectures for this task, struggle to effectively capture long-range dependencies due to the inherent locality of CNNs and the computational complexity of Transformers. To address this limitation, we introduce TTT-UNet, a novel framework that integrates Test-Time Training (TTT) layers into the traditional U-Net architecture for biomedical image segmentation. TTT-UNet dynamically adjusts model parameters during the test time, enhancing the model's ability to capture both local and long-range features. We evaluate TTT-UNet on multiple medical imaging datasets, including 3D abdominal organ segmentation in CT and MR images, instrument segmentation in endoscopy images, and cell segmentation in microscopy images. The results demonstrate that TTT-UNet consistently outperforms state-of-the-art CNN-based and Transformer-based segmentation models across all tasks. The code is available at https://github.com/rongzhou7/TTT-UNet

**Keywords:** Biomedical Image Segmentation, U-Net, Test-Time Training

## 1. Introduction

Accurate and reliable biomedical image segmentation is crucial for disease diagnosis, treatment planning, and clinical research, as it allows medical professionals to identify biological structures and measure their morphology (Qureshi et al., 2023; Cao et al., 2023). In recent years, convolutional neural networks (CNNs) (LeCun et al., 1995) have emerged as a promising approach in the field of biomedical image segmentation. Among various CNN-based techniques, U-Net (Ronneberger et al., 2015) stands out for its straightforward

---

[*] Contributed equally

structure and significant adaptability. Many enhancements and iterations (Huang et al., 2020; Cao et al., 2022; Zhou et al., 2018, 2019; Hatamizadeh et al., 2022, 2021) have been developed based on this U-shaped architecture, typically featuring a symmetric encoder-decoder design to capture multi-scale image features through convolutional operations. Leveraging this foundation, significant advancements have been achieved across a wide range of medical imaging applications (Yan et al., 2024; Zhang et al., 2024; Chen et al., 2024; Sun et al., 2024a; Zhou et al., 2023). These include cardiac segmentation in magnetic resonance (MR) imaging (Wang et al., 2021), multi-organ delineation in computed tomography (CT) scans (Li et al., 2018), and others (Safarov and Whangbo, 2021; Su et al., 2023).

Despite the remarkable representational capabilities of CNN-based models, their architectural design exhibits an inherent limitation in modeling long-range dependencies within images, because convolutional kernels are inherently local (Chen et al., 2023). While skip connections in the U-Net architecture facilitate the merging of low-level details with high-level features, they mainly serve to directly merge local features, which does not substantially boost the network's ability to model long-range dependencies. This limitation becomes particularly evident in biomedical imaging, where large variations in organ shapes, sizes, and textures are prevalent across different patients (Chen et al., 2023). Such variability poses challenges to the ability of CNN framework to consistently and accurately capture information across extended spatial contexts, highlighting the need for innovative approaches to address this fundamental constraint.

Recognizing the limitations of CNNs in capturing long-range dependencies, the research community has shifted interest towards Transformer models for their ability to naturally understand global contexts (Ji et al., 2021). This transition is evidenced in biomedical image segmentation, where approaches like TransUNet (Chen et al., 2021), UNETR (Hatamizadeh et al., 2022), SwinUNETR (Hatamizadeh et al., 2021) demonstrate the potential of integrating Transformers. These hybrid models that blend CNNs for high-resolution spatial detail and Transformers for global context emerge as an effective strategy.

Nevertheless, despite their ability to capture global dependencies, Transformers are computationally intensive (Vaswani et al., 2017), especially in dense biomedical image segmentation tasks. Mamba (Gu and Dao, 2023), a state-space model designed for efficient sequence modeling, offers a more computationally efficient approach to long-range dependency modeling. Building on this, U-Mamba (Ma et al., 2024a) integrates Mamba within U-Net, effectively combining high-resolution spatial detail with long-range dependency modeling to enhance biomedical image segmentation. Despite these advancements, U-Mamba and similar models, still face challenges in expressiveness, particularly over extended contexts, where their fixed-size hidden states limit their ability to capture complex and nuanced dependencies.

Recently, TTT (Test-Time Training) (Sun et al., 2024b) has emerged as a new class of sequence modeling layers with linear complexity and an expressive hidden state. TTT treats the traditional fixed hidden state as a machine learning model itself, which can be dynamically updated through self-supervised learning. This dynamic adjustment allows the model to refine its parameters based on test data, providing greater flexibility and expressiveness in capturing intricate long-range dependencies. Compared to Transformers and Mamba, TTT layers maintain efficiency and offer superior performance in handling long-context sequences.

In this paper, we introduce TTT-UNet, a novel hybrid architecture that incorporates TTT layers within the traditional U-Net framework to address the inherent limitations in

modeling long-range dependencies in biomedical image segmentation tasks. The TTT layers dynamically adapt its parameters during test time, allowing it to more effectively capture both localized details and long-range dependencies. Our extensive experiments across various medical imaging datasets demonstrate that TTT-UNet consistently outperforms existing state-of-the-art models. The results highlight the model's effectiveness in handling complex anatomical structures and its robustness in diverse clinical scenarios. Particularly, TTT-UNet has shown significant improvements in biomedical image segmentation tasks, making it a versatile solution for medical image analysis. Our contributions are summarized as follows:

- We introduce TTT-UNet, the first exploration of integrating Test-Time Training layers into U-Net architecture for medical image segmentation, which allows the model to perform self-supervised adaptation during test time. This hybrid design effectively explores the potential of TTT layers for modeling long-range dependencies and improves the model's generalization capability across diverse data distributions.

- We conduct comprehensive evaluations across diverse medical imaging datasets, including 3D abdominal organ segmentation in CT and MRI, instrument segmentation in endoscopy, and cell segmentation in microscopy images. TTT-UNet achieves consistent improvements over state-of-the-art models in 3D and 2D segmentation, demonstrating the feasibility of TTT mechanisms for biomedical image segmentation.

In summary, TTT-UNet represents a significant advancement in biomedical image segmentation, offering a robust and adaptable approach that leverages the strengths of CNNs and TTT layers. This work lays the foundation for future developments in adaptive and context-aware medical image analysis technologies.

## 2. Related work

### 2.1. U-Net and variants

CNN-based and Transformer-based models have significantly advanced the field of biomedical image segmentation. U-Net (Ronneberger et al., 2015), a representative among CNN-based approaches, features a symmetrical encoder-decoder architecture enhanced with skip connections to better preserve details. Various enhancements (Myronenko, 2019), such as the self-configuring nnU-Net (Isensee et al., 2021) framework, have been built on this U-shaped design, demonstrating robust performance across a variety of biomedical image segmentation challenges. For Transformer, TransUnet (Chen et al., 2021) stands out by integrating the Vision Transformer (ViT) (Dosovitskiy et al., 2020) for feature extraction in the encoding phase and coupling it with CNN for decoding, demonstrating its capability for processing global information. Swin-UNETR (Hatamizadeh et al., 2021) and UNETR (Hatamizadeh et al., 2022) blend Transformer architectures with traditional U-Net to enhance 3D imaging analysis. Additionally, Swin-UNet (Cao et al., 2022) delves into the use of Swin Vision Transformer blocks (Liu et al., 2021) within a U-Net framework, further expanding the exploration of Transformer technology in medical imaging.

## 2.2. Hybrid models

SSMs, such as Mamba, have recently gained prominence as a powerful component for developing deep networks, achieving cutting-edge performance in analyzing long-sequence data (Goel et al., 2022; Fu et al., 2022). In the realm of biomedical image segmentation, U-Mamba (Ma et al., 2024a) presents a novel SSM-CNN hybrid approach, signifying the first application of SSMs in the medical image domain. Further developments include SegMamba (Xing et al., 2024) and nnMamba (Gong et al., 2024), which combine SSMs in the encoder with CNNs in the decoder, illustrating the versatility and effectiveness of SSMs in enhancing medical imaging analysis.

## 3. Method

TTT-UNet follows the conventional U-Net structure, designed to effectively capture both local features and long-range dependencies. The network inherits the encoder-decoder design commonly used for segmentation tasks, with each stage contributing to effective multi-scale feature representation across layers. As shown in **Figure 1**, TTT-UNet integrates Test-Time Training (TTT) layers into the TTT building blocks within the U-Net network.

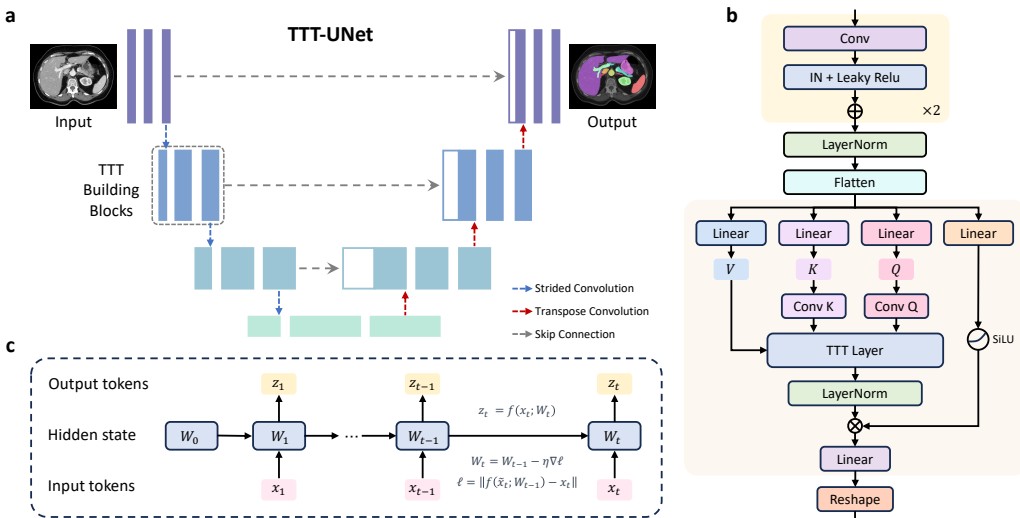

Figure 1: (a) Overview of TTT-UNet with test-time training layers integrated into the encoder and decoder. (b) TTT building block incorporating hidden-state adaptation. (c) Internal structure of the TTT layer with projection modules and the self-supervised update mechanism.

This integration enables the model to continuously update its parameters based on test data, enhancing its feature extraction capabilities in the encoder and allowing it to adaptively learn long-range dependencies. Subsequently, we introduce the TTT layer and then describe how it is integrated into the TTT building blocks within the U-Net architecture.

### 3.1. TTT layers

Traditional sequence modeling layers, such as RNNs, compress the context of a sequence $x_1, \ldots, x_t$ into a fixed-size hidden state $h_t$. For RNNs, the hidden state $h_t$ at time step $t$ is updated based on the current input $x_t$ and the previous hidden state $h_{t-1}$ through linear transformation matrices $\theta_h$ and $\theta_x$ and a non-linear activation function $\sigma$:

$$h_t = \sigma(\theta_h h_{t-1} + \theta_x x_t),$$

where $\theta_h$ and $\theta_x$ are learned parameters. The output $z_t$ is then generated from the hidden state: $z_t = \phi(h_t)$, where $\phi$ represents a linear or non-linear transformation.

However, the fixed size of the hidden state limits performance when dealing with long contexts due to its finite capacity to represent contextual information.

To address this limitation, a new class of sequence modeling layers, referred to as TTT layers (Sun et al., 2024b) is introduced, where the hidden state is treated as a trainable model and is updated through self-supervised learning.

Specifically, in a TTT layer (**Figure 1c**) , the hidden state $h_t$ at time step $t$ is treated as a trainable model $f$ with weights $W_t$, which is updated based on the current input $x_t$:

$$W_t = W_{t-1} - \eta \nabla \ell(W_{t-1}; x_t)$$

The output token $z_t$ is then generated using trainable model $f$ with weights $W_t$:

$$z_t = f(x_t; W_t)$$

In the basic naive version, the self-supervised loss $\ell$ aims to reconstruct the corrupted input $\tilde{x}_t$. This approach is straightforward and focuses on learning to recover the original input from its corrupted version:

$$\ell(W; x_t) = \|f(\tilde{x}_t; W) - x_t\|^2$$

While this naive reconstruction method is effective in certain scenarios, it has inherent limitations in capturing the complex dependencies within the input data, especially in tasks requiring a more nuanced understanding of the input context.

To address these limitations, we follow a more sophisticated self-supervised task that leverages multiple views of the input data. Instead of directly reconstructing the corrupted input, we introduce learnable matrices $\theta_K$ and $\theta_V$ to project the input into different views. We refer to these projections as Training View (K), Label View (V), and Test View (Q), which are used for self-supervised adaptation. The training view $K = \theta_K x_t$ captures the essential information needed for learning, while the label view $V = \theta_V x_t$ provides a target for reconstruction:

$$\ell(W; x_t) = \|f(\theta_K x_t; W) - \theta_V x_t\|^2$$

This approach allows the model to selectively focus on the most relevant features of the input, improving its ability to capture long-range dependencies and subtle relationships within the data. The output token $z_t$ is then generated as follow:

$$z_t = f(\theta_Q x_t; W_t),$$

where $f$ is a function parameterized by $W_t$, which can be a linear model or a MLP. Here, the projection $\theta_Q$ is used to obtain the test view $Q = \theta_Q x_t$, which introduces additional

flexibility by allowing the model to emphasize different aspects of the input data during inference. This approach enables the model to focus on the most informative features in the context of the current test case, thereby enhancing its ability to adapt to new, unseen data. TTT layers capture long-range dependencies by treating the hidden state as a trainable model updated sequentially. Each token updates the hidden parameters, allowing subsequent tokens to benefit from accumulated context, enabling global information propagation without quadratic attention complexity.

### 3.2. TTT-UNet architecture

The TTT-UNet architecture integrates the traditional U-Net structure with TTT layers, allowing the network to adapt during testing through self-supervised learning dynamically. The architecture is composed of an encoder-decoder structure, where the encoder is enhanced with TTT building blocks to improve adaptability, while the decoder follows the standard U-Net design focused on reconstructing the segmentation map.

**Encoder.** The encoder in TTT-UNet follows the traditional U-Net design, comprising multiple convolutional layers. These layers are interspersed with TTT building blocks, which are critical components that enable the model to adjust its parameters dynamically during test time. Each layer in the encoder progressively downscales the input image while capturing both local and long-range features essential for segmentation tasks. Including TTT building blocks within the encoder ensures the model can adapt to varying data distributions encountered during testing.

**TTT building blocks.** As illustrated in **Figure 1b**, the TTT building blocks are the core components that allow for the test-time adaptability of the model. Initially, the input features pass through two successive Residual blocks (He et al., 2016), each comprising a standard convolutional layer, followed by Instance Normalization (IN) (Ulyanov et al., 2016) and Leaky ReLU activation (Maas et al., 2013). Subsequently, the features are normalized using Layer Normalization (Ba et al., 2016), and flattened, making them suitable for linear transformations. For 2D inputs, we flatten $(B, C, H, W) \rightarrow (B, H \cdot W, C)$; for 3D inputs, $(B, C, D, H, W) \rightarrow (B, D \cdot H \cdot W, C)$. Then the flattened features undergo three separate linear transformation branches, obtaining the different features denoted as $V$, $K$, and $Q$ respectively. Additional convolutional operations (`Conv K` and `Conv Q`) are applied to the $K$ and $Q$ vectors, allowing the model to focus on specific aspects of the features during test-time training. Meanwhile, the fourth branch performs a linear transformation followed by a SiLU activation function (Hendrycks and Gimpel, 2016), further enriching the feature representations. Then the processed $V$, $K$, and $Q$ are fed into the TTT Layer, where self-supervised learning occurs. In this layer, the model dynamically updates its weights based on the self-supervised task applied to the processed $V$, $K$, and $Q$ vectors, as detailed in 3.1. The output from the TTT Layer is further normalized using Layer Normalization (Ba et al., 2016) before being passed on. Finally, this output and the fourth branch output mentioned before are combined via the Hadamard product, followed by a linear transformation and reshaping to fit the required dimensions for subsequent layers in Decoder.

**Decoder** The decoder in our model maintains the classic U-Net structure, integrating Residual blocks and transposed convolutions to enhance the capture of detailed local features and support resolution recovery. We also incorporate the skip connections in U-Net, ensuring

the effective transfer of hierarchical features from the encoder to the decoder. The final output of the decoder is refined through a $1\times1\times1$ convolutional layer and a Softmax activation, which generates the final segmentation probability map.

We implement two TTT-UNet variants: TTT-UNet_Bot, which applies TTT layers only in the bottleneck while retaining standard Residual blocks elsewhere, and TTT-UNet_Enc, which integrates TTT layers throughout the encoder for broader self-supervised adaptation.

## 4. Experiments

### 4.1. Datasets

To evaluate the performance and scalability of TTT-UNet, we utilize four biomedical image datasets across a variety of segmentation tasks and imaging modalities, including Abdomen CT dataset (Ma et al., 2024c), Abdomen MRI dataset (Ji et al., 2022), Endoscopy dataset (Allan et al., 2019) and Microscopy dataset (Ma et al., 2024b).

**Abdomen CT.** The Abdomen CT (Ma et al., 2023b) dataset, from the MICCAI 2022 FLARE challenge, includes the segmentation of 13 abdominal organs from 50 CT scans in both the training and testing sets. The organs segmented include the liver, spleen, pancreas, kidneys, stomach, gallbladder, esophagus, aorta, inferior vena cava, adrenal glands, and duodenum.

**Abdomen MRI.** The Abdomen MRI (Ji et al., 2022) dataset, from the MICCAI 2022 AMOS Challenge, focuses on the segmentation of the same 13 abdominal organs, using MRI scans. It consists of 60 MRI scans for training and 50 for testing. Additionally, we generate a 2D version of this dataset by converting the 3D abdominal MRI scans into 2D slices. This conversion enables us to evaluate TTT-UNet under the common 2D segmentation setting, which is widely used in practice due to its lower computational requirements. The conversion retains the same 13 organs, ensuring consistent evaluation across both 2D and 3D modalities.

**Endoscopy images.** From the MICCAI 2017 EndoVis Challenge (Allan et al., 2019), this dataset focuses on instrument segmentation within endoscopy images, featuring seven distinct instruments, including the large needle driver, prograsp forceps, monopolar curved scissors, cadiere forceps, bipolar forceps, vessel sealer, and a drop-in ultrasound probe. The dataset is split into 1800 training frames and 1200 testing frames.

**Microscopy images.** This dataset, from the NeurIPS 2022 Cell Segmentation Challenge (Ma et al., 2023a), is used for cell segmentation in microscopy images, consisting of 1000 training images and 101 testing images. Following U-Mamba (Ma et al., 2024a), we address this as a semantic segmentation task, focusing on cell boundaries and interiors rather than instance segmentation.

### 4.2. Experimental setup

The setting of our experiments is the same as that in U-Mamba (Ma et al., 2024a) and nnU-Net (Isensee et al., 2021) to ensure a fair comparison, as shown in **Table 6** We adopt an unweighted combination of Dice loss and cross-entropy loss for all datasets and utilize the SGD optimizer with an initial learning rate of 1e-2. The training duration for each dataset is set to 1000 epochs, conducted on a single NVIDIA A100 GPU. Leveraging the self-configuring capabilities from nnU-Net, the number of network blocks adjusts automatically according

to the dataset. For evaluation metrics, we employ the Dice Similarity Coefficient (DSC) and Normalized Surface Distance (NSD) to assess performance in abdominal multi-organ segmentation for MR scans, as well as instrument segmentation in Endoscopy images. For the cell segmentation task, we utilize the F1 score to evaluate method performance.

## 4.3. Baselines and metrics

In our evaluation of TTT-UNet, we compare against two prominent CNN-based segmentation methods: nnU-Net (Isensee et al., 2021) and SegResNet (Myronenko, 2019). Additionally, we include a comparison with UNETR (Hatamizadeh et al., 2022) and Swin-UNETR (Hatamizadeh et al., 2021), a Transformer-based network that has gained popularity in biomedical image segmentation tasks. U-Mamba (Ma et al., 2024a), a recent Mamba-based method, is also included in our comparison to provide a comprehensive overview of its performance. For each model, we implement their recommended optimizers to ensure consistency in training conditions. To maintain fairness across all comparisons, we apply the default image preprocessing in nnU-Net (Isensee et al., 2021).

## 4.4. Quantitative segmentation results

Table 1: Results summary of 2D segmentation tasks: organ segmentation in abdomen MRI scans, instruments segmentation in endoscopy images, and cell segmentation in microscopy images.

| Methods | Organs in Abdomen MRI | | Instruments in Endoscopy | | Cells in Microscopy |
|---|---|---|---|---|---|
| | DSC | NSD | DSC | NSD | F1 |
| nnU-Net | 0.7450±0.1117 | 0.8153±0.1145 | 0.6264±0.3024 | 0.6412±0.3074 | 0.5383±0.2657 |
| SegResNet | 0.7317±0.1379 | 0.8034±0.1386 | 0.5820±0.3268 | 0.5968±0.3303 | 0.5411±0.2633 |
| UNETR | 0.5747±0.1672 | 0.6309±0.1858 | 0.5017±0.3201 | 0.5168±0.3235 | 0.4357±0.2572 |
| SwinUNETR | 0.7028±0.1348 | 0.7669±0.1442 | 0.5528±0.3089 | 0.5683±0.3123 | 0.3967±0.2621 |
| U-Mamba_Bot | 0.7588±0.1051 | 0.8285±0.1074 | 0.6540±0.3008 | 0.6692±0.3050 | 0.5389±0.2817 |
| U-Mamba_Enc | 0.7625±0.1082 | 0.8327±0.1087 | 0.6303±0.3067 | 0.6451±0.3104 | 0.5607±0.2784 |
| TTT-UNet_Bot | **0.7750±0.1022** | **0.8452±0.1080** | **0.6643±0.3018** | **0.6799±0.3056** | **0.5818±0.2410** |
| TTT-UNet_Enc | **0.7725±0.1044** | **0.8540±0.1032** | **0.6696±0.3018** | **0.6820±0.3080** | **0.5773±0.2435** |

**Table 1** presents the 2D segmentation results across three datasets. TTT-UNet variants consistently achieve the best performance on all tasks. For organ segmentation in Abdomen MRI, TTT-UNet_Bot obtains the highest DSC (0.7750) while TTT-UNet_Enc achieves the best NSD (0.8540), both outperforming U-Mamba variants and other baselines by a notable margin. In the Endoscopy dataset, TTT-UNet_Enc achieves the highest DSC (0.6696) and NSD (0.6820), demonstrating its effectiveness in capturing fine-grained details of surgical instruments despite their small size and variable appearances. For cell segmentation in Microscopy images, TTT-UNet_Bot obtains the best F1 score (0.5818), showing robustness to the high variability and noise inherent in microscopy data.

**Table 2** presents the 3D segmentation results on Abdomen CT and MRI datasets. TTT-UNet_Bot achieves the highest DSC on both CT (0.8709) and MRI (0.8677), consistently outperforming U-Mamba_Bot and U-Mamba_Enc. On the MRI dataset, TTT-UNet_Bot also achieves the best NSD (0.9247), indicating superior boundary preservation. Notably,

Table 2: Results summary of 3D organ segmentation on abdomen CT and MRI datasets.

| Methods | Organs in Abdomen CT | | Organs in Abdomen MRI | |
|---|---|---|---|---|
| | DSC | NSD | DSC | NSD |
| nnU-Net | 0.8615±0.0790 | 0.8972±0.0824 | 0.8309±0.0769 | 0.8996±0.0729 |
| SegResNet | 0.7927±0.1162 | 0.8257±0.1194 | 0.8146±0.0959 | 0.8841±0.0917 |
| UNETR | 0.6824±0.1506 | 0.7004±0.1577 | 0.6867±0.1488 | 0.7440±0.1627 |
| SwinUNETR | 0.7594±0.1095 | 0.7663±0.1190 | 0.7565±0.1394 | 0.8218±0.1409 |
| U-Mamba_Bot | 0.8683±0.0808 | **0.9049±0.0821** | 0.8453±0.0673 | 0.9121±0.0634 |
| U-Mamba_Enc | 0.8638±0.0908 | 0.8980±0.0921 | 0.8501±0.0732 | 0.9171±0.0689 |
| TTT-UNet_Bot | **0.8709±0.1011** | 0.8995±0.0721 | **0.8677±0.0482** | **0.9247±0.0631** |

TTT-UNet exhibits lower variance across all tasks compared to other methods, suggesting more stable and reliable predictions across diverse test samples. To validate statistical significance, we conducted paired Wilcoxon signed-rank tests on the 3D AbdomenMRI dataset ($n$=60). The results show that TTT-UNet significantly outperforms nnU-Net ($p$=0.016).

These consistent improvements across different modalities (CT, MRI, endoscopy, microscopy) and dimensions (2D, 3D) demonstrate the effectiveness of TTT layers. The learnable hidden state in TTT, which updates dynamically via self-supervised learning while processing the input sequence, provides richer representational capacity, enabling better modeling of complex anatomical structures and varying imaging conditions.

### 4.5. Qualitative segmentation results

As shown in **Figure 2**, the segmentation results on the Abdomen MRI dataset demonstrate TTT-UNet's effectiveness in handling complex anatomical structures. The predictions show strong alignment with ground truth, particularly in regions with significant anatomical variability, highlighting robust performance in segmenting intricate abdominal organs.

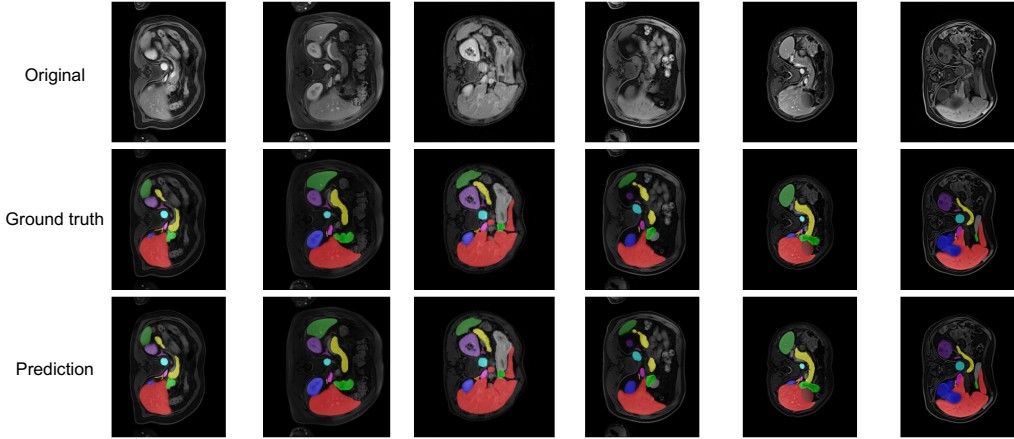

Figure 2: The visualization results of TTT-UNet on Abdomen MRI datasets.

**Figure 3** provides further insights through the segmentation results on the Endoscopy and Microscopy datasets. In the Endoscopy dataset, TTT-UNet successfully delineates the surgical instruments, which are challenging due to their small size and diverse appearances. This capability underlines the model's strength in capturing fine details and its adaptability to various shapes and textures. Similarly, in the Microscopy dataset, TTT-UNet demonstrates

Table 3: Computational cost and performance for 2D segmentation tasks. DSC is reported for AbdomenMRI and Endoscopy; F1-score for Microscopy.

| Dataset | Model | Params (M) | FLOPs (G) | Memory (MB) | Time (ms) | DSC/F1 |
|---|---|---|---|---|---|---|
| AbdomenMRI | nnU-Net | 5.69 | 18.14 | 125.0 | 4.40±0.03 | 0.745±0.112 |
| | SegResNet | 6.30 | 24.49 | 193.6 | 5.28±0.01 | 0.732±0.138 |
| | UNETR | 115.77 | 42.14 | 634.8 | 10.80±0.04 | 0.575±0.167 |
| | SwinUNETR | 25.14 | 27.88 | 294.3 | 11.02±0.49 | 0.703±0.135 |
| | U-Mamba_Bot | 8.26 | 33.65 | 198.8 | 6.90±0.20 | 0.759±0.105 |
| | U-Mamba_Enc | 8.39 | 34.51 | 245.0 | 7.30±0.22 | 0.763±0.108 |
| | TTT-UNet (ours) | 8.01 | 33.69 | 197.4 | 13.22±1.24 | **0.775±0.102** |
| Endoscopy | nnU-Net | 5.69 | 43.63 | 256.1 | 12.86±0.05 | 0.626±0.302 |
| | SegResNet | 6.30 | 58.88 | 222.2 | 12.60±0.04 | 0.582±0.327 |
| | UNETR | 116.59 | 111.47 | 662.4 | 21.42±0.06 | 0.502±0.320 |
| | SwinUNETR | 25.14 | 67.12 | 450.6 | 18.85±0.04 | 0.553±0.309 |
| | U-Mamba_Bot | 8.26 | 80.88 | 296.0 | 9.04±0.05 | 0.654±0.301 |
| | U-Mamba_Enc | 8.39 | 82.94 | 526.7 | 15.40±0.64 | 0.630±0.307 |
| | TTT-UNet (ours) | 8.01 | 80.96 | 294.6 | 18.42±1.52 | **0.664±0.302** |
| Microscopy | nnU-Net | 5.69 | 46.49 | 272.4 | 11.64±0.71 | 0.538±0.266 |
| | SegResNet | 6.30 | 62.76 | 191.3 | 13.57±0.06 | 0.541±0.263 |
| | UNETR | 116.64 | 120.09 | 640.5 | 22.92±0.01 | 0.436±0.257 |
| | SwinUNETR | 25.14 | 71.74 | 465.6 | 19.98±0.05 | 0.397±0.262 |
| | U-Mamba_Bot | 8.26 | 86.23 | 305.1 | 9.47±0.02 | 0.539±0.282 |
| | U-Mamba_Enc | 8.39 | 88.43 | 558.8 | 16.27±0.03 | 0.561±0.278 |
| | TTT-UNet (ours) | 8.01 | 86.32 | 303.1 | 17.34±0.74 | **0.582±0.241** |

its robustness by accurately segmenting cell boundaries and interiors, even amidst high variability and noise levels. The model's performance in these diverse settings highlights its versatility and reliability across different medical imaging modalities.

The visual evidence presented in **Figures 2** and **3** aligns with the quantitative improvements reported in Table 1. These results underscore the consistent ability of TTT-UNet to deliver high-quality segmentations across diverse biomedical imaging modalities.

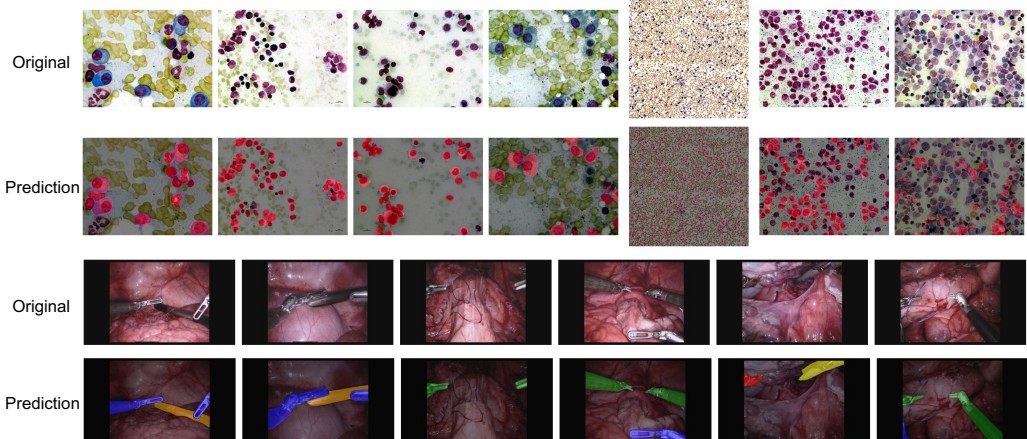

Figure 3: Visualization results of TTT-UNet on Microscopy and Endoscopy datasets.

Table 4: Computational cost and performance for 3D segmentation tasks.

| Dataset | Model | Params (M) | FLOPs (G) | Memory (MB) | Time (ms) | DSC |
|---|---|---|---|---|---|---|
| | nnU-Net | 16.55 | 623.66 | 2286.1 | 39.21±0.18 | 0.862±0.079 |
| | SegResNet | 18.80 | 764.19 | 1881.1 | 114.48±0.08 | 0.793±0.116 |
| | UNETR | 121.47 | 234.87 | 1799.2 | 68.05±0.07 | 0.682±0.151 |
| AbdomenCT | SwinUNETR | 15.70 | 260.50 | 3949.5 | 144.55±0.14 | 0.759±0.110 |
| | U-Mamba_Bot | 22.58 | 1345.84 | 2316.9 | 94.27±0.11 | 0.868±0.081 |
| | U-Mamba_Enc | 23.17 | 1367.57 | 5516.6 | 178.04±0.15 | 0.864±0.091 |
| | TTT-UNet (ours) | 22.32 | 1345.90 | 2315.6 | 97.11±0.03 | **0.871±0.101** |
| | nnU-Net | 16.55 | 519.72 | 1915.3 | 32.78±0.12 | 0.831±0.077 |
| | SegResNet | 18.80 | 636.82 | 1581.2 | 95.95±0.04 | 0.815±0.096 |
| | UNETR | 121.39 | 194.57 | 1580.3 | 56.41±0.04 | 0.687±0.149 |
| AbdomenMRI | SwinUNETR | 15.70 | 217.41 | 3391.3 | 121.46±0.11 | 0.757±0.139 |
| | U-Mamba_Bot | 22.58 | 1121.54 | 1947.0 | 79.13±0.11 | 0.845±0.067 |
| | U-Mamba_Enc | 23.17 | 1139.64 | 4613.3 | 148.74±0.15 | 0.850±0.073 |
| | TTT-UNet (ours) | 22.32 | 1121.58 | 1945.3 | 81.66±0.20 | **0.868±0.048** |

## 4.6. Computational cost analysis

To evaluate the computational efficiency of TTT-UNet, we conduct a comprehensive analysis measuring four metrics: parameter count (M), FLOPs (G), peak GPU memory (MB), and inference time (ms) averaged over 10 runs with standard deviation. All experiments are conducted on a single NVIDIA A100 80G GPU with PyTorch 2.2.2 and CUDA 12.1. For computational analysis, we report TTT-UNet with TTT layers applied at the bottleneck (TTT-UNet_Bot), which we consider a balanced choice for deployment, especially in 3D settings. Tables 3 and 4 report the computational cost for 2D and 3D segmentation tasks, respectively. For 3D segmentation tasks, TTT-UNet achieves nearly identical computational cost to U-Mamba_Bot with only ∼3% additional inference time (97.11ms vs 94.27ms on CT, 81.66ms vs 79.13ms on MRI) and equivalent memory footprint (∼2316MB on CT, ∼1946MB on MRI), while delivering +0.3% DSC on CT and +2.3% DSC on MRI. The additional overhead primarily comes from the gradient computation within the TTT layers during inference, where each TTT layer performs a single lightweight self-supervised update. TTT-UNet also achieves the lowest variance on 3D AbdomenMRI (std=0.048 vs 0.067–0.077 for baselines), indicating more stable predictions across diverse test samples. For 2D segmentation tasks, TTT-UNet requires approximately 2× inference time compared to U-Mamba_Bot, but remains faster than UNETR and comparable to SwinUNETR. Importantly, TTT-UNet maintains a similar memory footprint to U-Mamba_Bot and achieves the smallest parameter count (8.01M) among all compared methods with consistent accuracy improvements.

## 5. Discussion and conclusion

The experimental results from multiple biomedical image segmentation tasks consistently demonstrate that TTT-UNet achieves notable improvements over a range of state-of-the-art methods. A key factor contributing to this improvement is the integration of TTT layers, which enable the model to dynamically adapt to the distinct characteristics and underlying data distribution of each test image. This capability leads to enhanced generalization,

especially in tasks involving diverse and complex imaging modalities, such as 3D abdomen CT, abdomen MRI, endoscopy, and microscopy datasets.

Furthermore, TTT-UNet's superior performance in handling high anatomical variability and complex spatial structures positions it as a robust tool for clinical applications. For both large-scale anatomical structures and smaller, intricate features, TTT-UNet has demonstrated the ability to deliver accurate segmentation results. This versatility is crucial in clinical scenarios where precision and adaptability are essential for effective diagnosis and treatment. The lower variance observed on 3D AbdomenMRI (std=0.048 vs 0.067–0.077 for baselines) suggests that TTT-UNet may particularly benefit organs with high anatomical variability, such as the pancreas, gallbladder, and adrenal glands, though detailed per-organ analysis is left for future work.

One of the primary advantages of TTT-UNet lies in its capacity to dynamically adjust model parameters during the test phase, which significantly enhances segmentation accuracy. Additionally, the lower variance in performance across different datasets emphasizes the model's robustness and consistency. Regarding test-time overfitting, TTT-UNet incorporates built-in safeguards: the TTT layer performs only a single gradient step per token rather than iterative optimization, and weights are reset to trained values for each new test sample (Appendix B.4). The learnable learning rate $\eta$ further allows the model to apply conservative, input-dependent updates, preventing excessive adaptation to noisy or outlier samples.

For clinical deployment, TTT layers are stateless across samples and do not introduce additional privacy risks. The hidden states are re-initialized for each input, no patient information persists across samples, and test-time updates operate only on internal feature representations rather than raw patient data.

**Limitations and future directions.** While TTT-UNet demonstrates consistent improvements, several limitations should be acknowledged. First, the computational overhead of TTT layers, though modest for 3D tasks (∼3% additional inference time), increases to approximately 2× for 2D tasks compared to U-Mamba, which may limit real-time applications such as intraoperative guidance and video-based surgical navigation. Second, the current analysis focuses on linear TTT layers; extending the theoretical analysis to non-linear variants (e.g., MLP-based) remains an open question. Third, while the key hyperparameters of TTT layers follow the original TTT design, a systematic sensitivity analysis specifically for medical image segmentation could further strengthen the understanding of the method.

Future work should focus on: (1) improving the efficiency of TTT layer implementations through gradient optimization and kernel fusion; (2) exploring input-dependent strategies that dynamically determine when test-time adaptation is most beneficial; (3) evaluating TTT-UNet on additional dense prediction tasks beyond medical segmentation, such as depth estimation and scene parsing; and (4) investigating the combination of TTT layers with pre-trained foundation models for medical imaging.

In conclusion, TTT-UNet represents an important step forward in biomedical image segmentation by offering a flexible and adaptive solution. Its ability to consistently outperform other models in both 2D and 3D segmentation tasks reinforces its potential as a reliable model for medical image analysis. As the model evolves, further optimization of test-time adaptation strategies, along with integration with large-scale, diverse clinical and specialized datasets, will pave the way for broader clinical adoption and deployment.

## 6. Acknowledgements

This work is partially supported by the National Science Foundation grants (MRI-2215789, IIS-2319451, CRII-2246067, ATD-2427915, POSE-2346158, POSE-2449280), National Institutes of Health grants (R01LM013519, RF1AG077820, R21EY034179), Department of Energy (DE-SC0025801), and Lehigh's grants under CORE, FRG and RIG.

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

## Appendix

### A. Dataset information

To comprehensively evaluate the performance and generalization ability of TTT-UNet across diverse biomedical imaging scenarios, we conduct experiments on four widely used datasets spanning both 2D and 3D segmentation tasks. These datasets cover abdominal CT and MRI scans, endoscopic surgical scenes, and microscopy cell images, providing a broad range of anatomical structures, imaging modalities, and domain shifts.

Table 5 summarizes the key characteristics of each dataset, including their dimensionality, number of training and testing samples, and segmentation targets. The abdominal multi-organ datasets (CT and MRI) contain volumetric scans with substantial anatomical variability, while the Endoscopy dataset features challenging surgical scenes with small, deformable instruments. The Microscopy dataset consists of fine-grained cellular structures with high appearance variability.

Table 5: Dataset information for segmentation tasks in biomedical imaging.

| Dataset | Dimension | #Training | #Testing | #Targets |
|---|---|---|---|---|
| Abdomen CT | 3D | 50 (4794 slices) | 50 (10894 slices) | 13 |
| Abdomen MRI | 3D | 60 (5615 slices) | 50 (3357 slices) | 13 |
| Endoscopy images | 2D | 1800 | 1200 | 7 |
| Microscopy images | 2D | 1000 | 101 | 2 |

To ensure fair and task-appropriate evaluation, TTT-UNet is configured with dataset-specific architectural and training settings. These include the patch size, batch size, number of stages, and pooling depth along each spatial axis. As shown in **Table 6**, 3D datasets use volumetric patches and smaller batch sizes due to memory constraints, while 2D datasets allow larger patch sizes and batches. The number of stages and pooling operations is chosen to balance receptive field size with model capacity.

Table 6: TTT-UNet configurations for each dataset.

| Dataset | Patch Size | Batch Size | #Stages | #Pooling per Axis |
|---|---|---|---|---|
| Abdomen CT | (40, 224, 192) | 2 | 6 | (3, 3, 5) |
| 3D Abdomen MR | (48, 160, 224) | 2 | 6 | (3, 5, 5) |
| 2D Abdomen MR | (320, 320) | 30 | 7 | (6, 6) |
| Endoscopy | (384, 640) | 13 | 7 | (6, 6) |
| Microscopy | (512, 512) | 12 | 8 | (7, 7) |

These dataset characteristics and configuration settings establish a comprehensive and diverse evaluation environment, enabling a rigorous assessment of TTT-UNet across varying imaging modalities, spatial resolutions, and task complexities.

## B. Training configuration

### B.1 Learning rate and optimizer

The training process adopts a poly learning rate schedule with an initial learning rate set to 0.01, which gradually decays following the equation:

$$\text{Learning rate} = \text{initial\_lr} \times \left( 1 - \frac{\text{epoch}}{\text{max\_epochs}} \right)^{0.9}$$

This decay schedule stabilizes the training by reducing the learning rate as training progresses. The optimizer used is Stochastic Gradient Descent (SGD) with Nesterov momentum set to 0.99, which aids in faster convergence and better optimization stability.

### B.2 Loss function

A combined loss function is employed to balance region overlap accuracy and pixel-wise classification accuracy. The total loss $\mathcal{L}$ is defined as:

$$\mathcal{L} = \text{Dice Loss} + \text{Cross-Entropy Loss}$$

where both components are weighted equally (weight=1). Deep supervision is applied by incorporating intermediate outputs from the decoder during training, with weights decreasing exponentially as $w_i = 1/2^i$ for shallower outputs, where the final (lowest resolution) output has weight 0.

### B.3 Data augmentation

We adopt nnU-Net's data augmentation pipeline, which applies the following transformations stochastically during training:

- **Geometric transformations**: Random rotations ($\pm 15°$), axis mirroring, scaling (0.7-1.4), and elastic deformations.

- **Intensity transformations**: Brightness and contrast adjustments, gamma correction ($\gamma \in [0.7, 1.5]$), and Gaussian noise ($\sigma \sim U(0, 0.1)$).

- **Simulation of acquisition artifacts**: Gaussian blur and low-resolution downsampling simulation.

These augmentations are applied on-the-fly during training to enhance model robustness without requiring additional storage.

### B.4 Test-Time Training configuration

During inference, TTT layers update their parameters through self-supervised learning on each test sample. The key hyperparameters are:

- **Mini-batch size**: 64 tokens per mini-batch for sequential parameter updates.

- **TTT learning rate**: Base learning rate $\eta_{\text{base}} = 1.0$, scaled by head dimension as $\eta = \eta_{\text{base}}/d_{\text{head}}$.

- **Update iterations**: One gradient step per mini-batch (no multiple iterations).

- **Weight initialization**: TTT layer weights are reset to their trained values for each new test sample, ensuring independence across samples.

For baseline methods (nnU-Net, U-Mamba, etc.), standard inference without test-time adaptation is used.

## C. Evaluation metrics

To assess the performance of TTT-UNet and its baseline models, we use several widely adopted evaluation metrics in biomedical image segmentation, ensuring a comprehensive analysis of segmentation accuracy and boundary preservation.

### C.1 Dice similarity coefficient (DSC)

The Dice Similarity Coefficient (DSC) is a widely used metric for evaluating the overlap between the predicted segmentation and the ground truth. It is defined as:

$$\text{DSC} = \frac{2|X \cap Y|}{|X| + |Y|}$$

where $X$ is the set of predicted pixels and $Y$ is the set of ground truth pixels. The DSC ranges from 0 to 1, with higher values indicating better segmentation performance. This metric is particularly useful for tasks where the accurate localization of organs or regions is important.

### C.2 Normalized surface distance (NSD)

The Normalized Surface Distance (NSD) measures the distance between the surfaces of the predicted and ground truth segmentations, normalized by the object's size. It is defined as:

$$\text{NSD} = \frac{1}{|S|} \sum_{p \in S} \min_{q \in G} d(p, q) \leq \tau$$

where $S$ and $G$ are the surfaces of the predicted segmentation and ground truth, and $d(p, q)$ is the Euclidean distance between points $p$ and $q$. The threshold $\tau$ defines the acceptable tolerance for boundary differences. NSD is important for ensuring the preservation of organ shapes and boundaries, especially in tasks involving complex anatomical structures.

### C.3 F1-Score

For binary classification problems (e.g., cell segmentation in microscopy images), we evaluate the performance using the F1-Score, which is defined as:

$$\text{F1} = 2 \times \frac{\text{Precision} \times \text{Recall}}{\text{Precision} + \text{Recall}}$$

This metric provides a balance between precision (avoiding false positives) and recall (capturing true positives) and is particularly useful in scenarios where class imbalance exists.

These metrics collectively provide a comprehensive evaluation of the model's performance across various biomedical imaging tasks, ensuring both regional overlap and boundary accuracy are properly assessed.

## D. Theoretical analysis and proofs

### D.1 CONVERGENCE ANALYSIS OF TTT LAYER PARAMETER UPDATES

In TTT-UNet, the parameter updates of the TTT layer follow the online gradient descent rule:

$$W_t = W_{t-1} - \eta \nabla \ell(W_{t-1}, x_t),$$

where $\ell(W; x_t) = \|f(\theta_K x_t; W) - \theta_V x_t\|^2$ is the self-supervised loss function, $\eta$ is the learning rate. We assume that $f$ is a linear model (i.e., $f(x; W) = Wx$) and analyze the convergence of the parameter updates.

---

*Theorem 1 Convergence of Online Gradient Descent:*
Assume the loss function $\ell(W; x)$ is strongly convex with respect to $W$, $L$-smooth, and the gradient of the input sequence $\{x_t\}$ satisfies $\|\nabla \ell(W; x_t)\| \leq G$.
If the learning rate $\eta \leq \frac{1}{L}$, the cumulative regret (as defined in online convex optimization) satisfies:

$$\sum_{t=1}^{T} \ell(W_t; x_t) - \min_W \sum_{t=1}^{T} \ell(W; x_t)$$
$$\leq \frac{\|W_1 - W^*\|^2}{2\eta} + \frac{\eta G^2 T}{2},$$

where $W^*$ is the optimal parameter. When $\eta = O(1/\sqrt{T})$, the average regret converges at a rate of $\eta = O(1/\sqrt{T})$.

---

**Proof:**

1. Assumptions of Strong Convexity and Smoothness:
   For the linear model $f(x; W) = Wx$, the loss function can be written as:

   $$\ell(W; x_t) = \|W\theta_K x_t - \theta_V x_t\|^2 = \|WK_t - V_t\|^2,$$

   where $K_t = \theta_K x_t$ and $V_t = \theta_V x_t$.

   The gradient is given by:

   $$\nabla \ell(W; x_t) = 2(WK_t - V_t)K_t^\top.$$

   If the input data satisfies $K_t K_t^\top \preceq LI$, then $\ell(W; x_t)$ is $L$-smooth. If $K_t K_t^\top \succeq \mu I$, then $\ell(W; x_t)$ is $\mu$-strongly convex.

2. Regret Bound:
   Based on the regret bound of online gradient descent (Zinkevich, 2003), for any $W^*$, we have:
   $$\sum_{t=1}^{T} \ell(W_t; x_t) - \sum_{t=1}^{T} \ell(W^*; x_t) \leq \frac{\|W_1 - W^*\|^2}{2\eta} + \frac{\eta G^2 T}{2}.$$

By choosing $\eta = \frac{1}{L\sqrt{T}}$, the average regret rate is $O(1/\sqrt{T})$, which shows that the sequence $\{W_t\}$ converges to the optimal parameter $W^*$.

## D.2 Multi-head projection mechanism

In TTT-UNet, the projection layers parameterized by $\theta_k$, $\theta_v$, and $\theta_q$ map the input features to a latent space. The self-supervised loss is defined as:

$$\mathcal{L}(W; x_t) = \|f_{\theta_k}(x_t; W) - \theta_v x_t\|_2^2$$

The projection mechanism is designed to dynamically adapt the model to test samples by capturing essential feature representations while minimizing the discrepancy between the predicted and target projections:

- The projection $K_t = \theta_k x_t$ extracts task-relevant feature representations, which are optimized during test time through TTT updates.

- The target projection $V_t = \theta_v x_t$ serves as a reference for minimizing the self-supervised loss.

- This mechanism ensures that the dynamically updated parameters $W_t$ learn to adapt the feature extraction process to unseen data distributions, effectively mitigating domain shifts.

This mechanism enables the TTT layer to handle domain shifts effectively, providing robust segmentation performance across diverse test samples.

## E. Visualizations of additional segmentation results

We present additional visualizations of segmentation results to demonstrate further the effectiveness of TTT-UNet across diverse medical imaging tasks and modalities. As shown in **Figures 4, 5, 6**, the model consistently performs well in capturing complex anatomical structures, fine-grained details, and variable features. **Figure 4** highlights the segmentation results on the Abdomen MRI dataset, where the model accurately delineates organ boundaries even in cases with significant anatomical variations. **Figure 5** showcases the model's robustness in segmenting cells within microscopy images, effectively handling ambiguous and highly variable cell boundaries. In **Figure 6**, we visualize the segmentation of surgical instruments in endoscopy images, demonstrating the model's ability to adapt to small, diverse instrument shapes and challenging environments. These visualizations further validate the model's capacity to generalize and maintain high performance in various clinical scenarios.

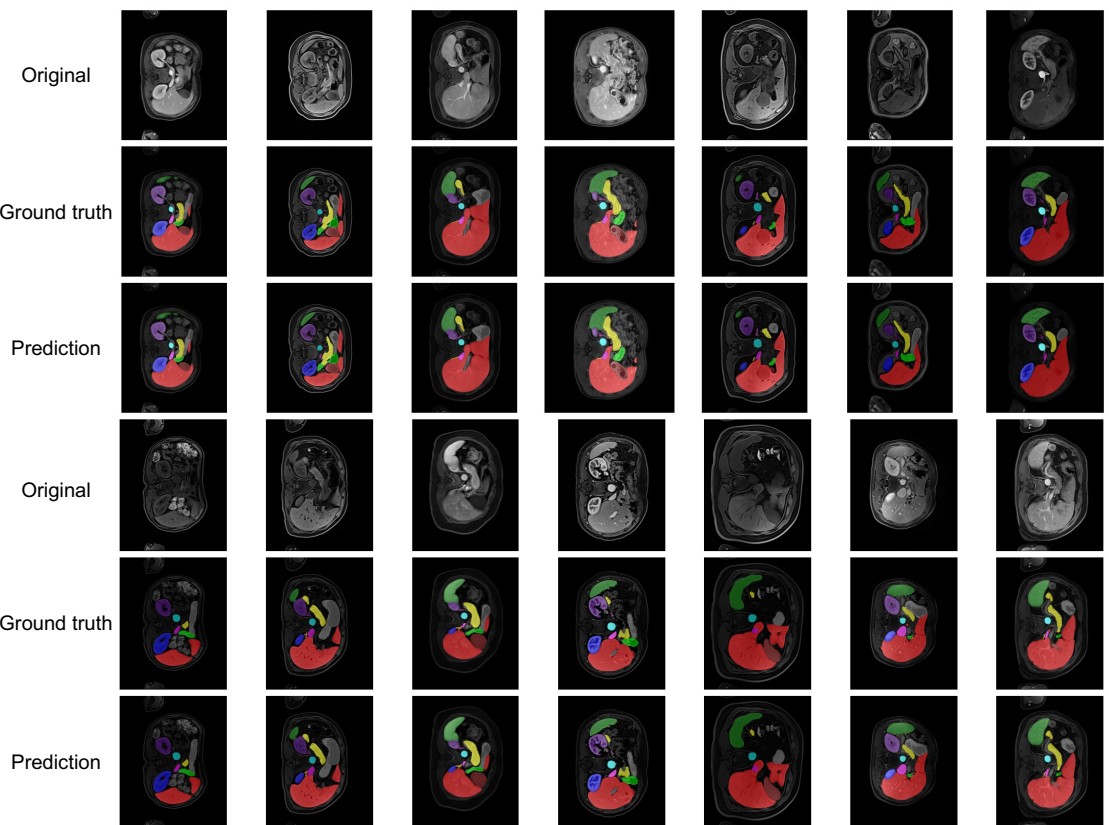

Figure 4: The visualization results of TTT-UNet on Abdomen MRI datasets.

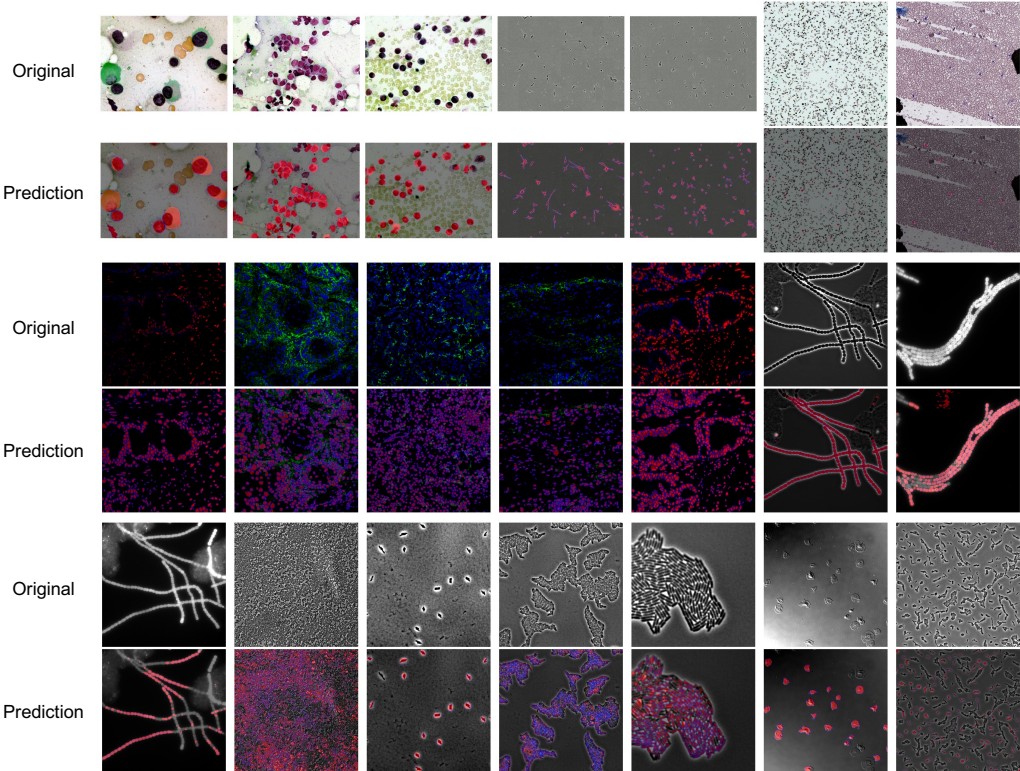

Figure 5: Visualization results of TTT-UNet on Microscopy dataset.

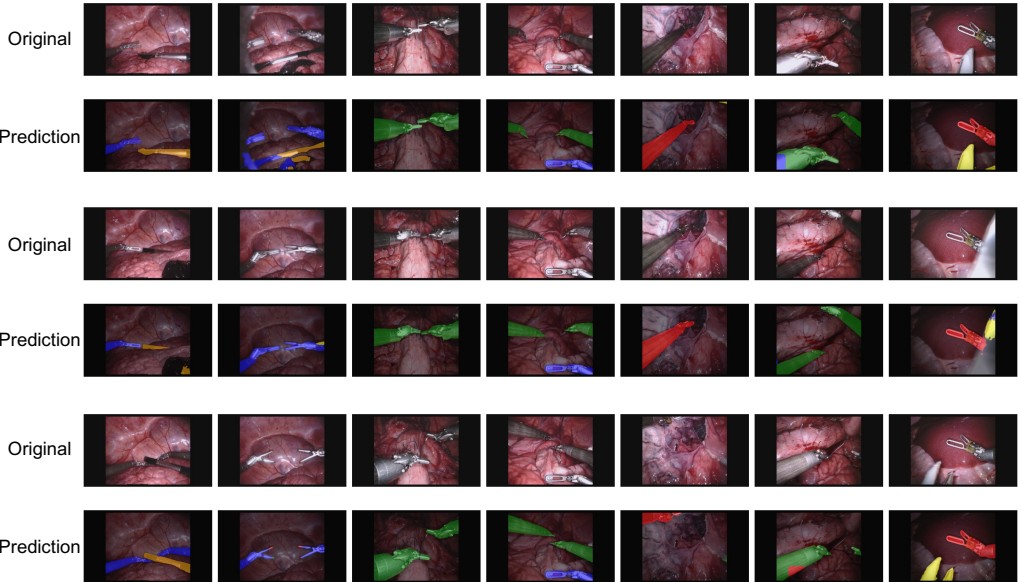

Figure 6: Visualization results of TTT-UNet on Endoscopy dataset.

