# OpenReview forum: "TTT-UNet: Enhancing U-Net with Test-Time Training Layers for Biomedical Image Segmentation"
_MIDL.io/2026/Conference — MIDL 2026 Poster_

### Official Review · Reviewer_Jyb7 · 2026-01-10

**Confidence:** 4
**Preliminary Rating:** 4
**Final Rating:** 5

**Summary:**

The paper introduces TTT-UNet, a hybrid architecture that integrates Test-Time Training layers into the U-Net framework to improve biomedical image segmentation. The authors aimed to enhance the model's ability to capture both local and long-range dependencies by dynamically adjusting its parameters during testing through self-supervised learning via adaptation of TTT layers. Experiments conducted across diverse biomedical imaging tasks demonstrated that the proposed architecture outperforms state-of-the-art models in segmentation accuracy, potentially providing more reliable and accurate image analysis.

**Strengths:**

The paper presents a well-organized and clearly outlined description of the proposed TTT-UNet model architecture, leveraging Test-Time Training layers for improved biomedical image segmentation. The comprehensive validation of the model across diverse medical imaging sources is well-noted, including MRI, endoscopy, and microscopy images, demonstrating the robustness of the proposed method. The model consistently outperforms multiple baseline and state-of-the-art methods in segmentation accuracy, highlighting its reliability and potential impact on diverse clinical tasks.

**Weaknesses:**

The paper is based on previously published work on TTT layers. Although the authors adapted the method for improved self-supervised tasks, they did not compare their approach with the original TTT method. Additionally, while the work is motivated by the need to enhance model capacity in capturing both local and long-range features, the authors mention this aspect without providing detailed elaboration on how the proposed method specifically addresses this need.

**Detailed Comments:**

- It is not quite clear from the paper how the proposed method "improves its ability to capture long-range dependencies and subtle relationships within the data".
- Persume U-Mamba_Bot and U-Mamba_Enc also included the TTT layers?

**Justification Of Final Rating:**

Thank you to the authors for the detailed rebuttal. As aforementioned, I believe this work would be of interest to the MIDL community, given its novelty and consistent performance improvement across various medical imaging modalities. Since the authors have addressed my main concerns, including justification of TTT layer adaptation and runtime analysis, I am adjusting the rating to strong accept.

**Justification Of The Preliminary Rating:**

The paper is well-organized and presents an interesting architecture that utilizes Test-Time Training layers in the UNet architecture for improved biomedical image segmentation. The method demonstrates consistent improvements in segmentation accuracy across various medical imaging modalities, such as MRI, endoscopy, and microscopy. Thus, this work could be of interest to the medical imaging community.

**Questions To Address In The Rebuttal:**

- Please elaborate on how the proposed adjustment to the original TTT layer method meets the need to capture local and long-range features and how the proposed adjustment may be more advantageous compared to the previously published TTT method.
- Stratifying the results by organs may better demonstrate the advantage of the proposed architecture in capturing nuanced features and context, especially for odd-shaped organs.
- Please comment on (or if possible, provide runtime analysis on) how the addition of the TTT architecture may impact computational cost, as compared to the baseline methods.

---

> ### Author Response · Authors · 2026-01-25
>
> **1. How TTT Captures Long-Range Dependencies**
> >[W] "While the work is motivated by the need to enhance model capacity in capturing both local and long-range features, the authors mention this aspect without providing detailed elaboration on how the proposed method specifically addresses this need."
> [Q1] "Please elaborate on how the proposed adjustment to the original TTT layer method meets the need to capture local and long-range features and how the proposed adjustment may be more advantageous compared to the previously published TTT method."
>
> **Response**
> We thank the reviewer for highlighting this core concern. The main limitation of standard CNN-based U-Net variants is that feature interactions are dominated by local receptive fields (even with skip connections), making it difficult to capture global context and subtle long-range relationships, particularly under distribution shifts [1,2].
>
> TTT layers address this limitation by treating the hidden state as a trainable model that is dynamically updated during inference [3]. After convolutional feature extraction, we flatten intermediate feature maps into tokens and process them within the TTT layer using an online update formulation. For each token, a lightweight self-supervised update is applied to the hidden model parameters, and these updated parameters are then used to process subsequent tokens. As a result, the representation of later tokens is influenced by the accumulated updates induced by earlier tokens, enabling information to propagate across the entire token sequence. This provides a mechanism for modeling global dependencies without relying on explicit quadratic attention, similar in spirit to recurrent models that propagate information through a shared hidden state [4], as well as more recent state-space models designed for long-range sequence modeling [5].
>
> Compared to attention-based Transformers, TTT does not compute pairwise token-to-token interactions [6]. Instead, it summarizes contextual information through a compact, learnable hidden model that is adaptively updated at test time [3]. This provides an alternative to fixed attention patterns by using a sample-specific adaptive state, which is well suited to domain shifts commonly observed in biomedical imaging.
>
> Compared to the previously published TTT formulation [3], our adjustments are advantageous for dense segmentation in several ways. First, we integrate TTT layers into a U-Net backbone rather than using TTT as a standalone sequence model. This preserves strong local inductive bias via convolutions and skip connections, while TTT layers provide global, sample-specific adaptation over tokenized features. Second, we introduce multi-view projections (training, label, and test views) to decouple the self-supervised adaptation objective from the prediction pathway, making the test-time updates more task-aligned for dense prediction rather than naive reconstruction. Third, we tailor the tokenization and placement of TTT layers to 2D and 3D feature maps at selected network stages, which is more appropriate for segmentation than directly applying the original TTT formulation on raw inputs. Together, these design choices support effective modeling of both local and long-range features in biomedical image segmentation.
>
> We will clarify this mechanism and the distinction from attention-based blocks more explicitly in the revised manuscript.
>
> *[1] Long, Jonathan, Evan Shelhamer, and Trevor Darrell. "Fully convolutional networks for semantic segmentation." Proceedings of the IEEE conference on computer vision and pattern recognition. 2015.*
>
>  *[2] Ronneberger, Olaf, Philipp Fischer, and Thomas Brox. "U-net: Convolutional networks for biomedical image segmentation." International Conference on Medical image computing and computer-assisted intervention. Cham: Springer international publishing, 2015.*
>
> *[3] Sun, Yu, et al. "Learning to (Learn at Test Time): RNNs with Expressive Hidden States." Proceedings of the 42nd International Conference on Machine Learning, 2025.*
>
> *[4] Graves, Alex. "Long short-term memory." Supervised sequence labelling with recurrent neural networks (2012): 37-45.*
>
> *[5] Gu, Albert, Karan Goel, and Christopher Ré. " Efficiently Modeling Long Sequences with Structured State Spaces." Proceedings of the International Conference on Learning Representations, 2022.*
>
>  *[6] Vaswani, Ashish, et al. "Attention is all you need." Advances in neural information processing systems 30 (2017).*

---

> > ### Author Response · Authors · 2026-01-25
> >
> > **2. Comparison with Original TTT Method**
> > >[W] "The paper is based on previously published work on TTT layers. Although the authors adapted the method for improved self-supervised tasks, they did not compare their approach with the original TTT method."
> > [Q1] "Please elaborate how the proposed adjustment may be more advantageous compared to the previously published TTT method."
> >
> > **Response**
> >
> > We thank the reviewer for the question regarding the comparison with the original TTT method. The original TTT method was designed for sequence modeling in natural language processing, where inputs are one-dimensional token sequences. In contrast, biomedical image segmentation involves dense 2D or 3D spatial structures and requires jointly modeling fine-grained local details and global context.
> >
> > Our work adapts the original TTT formulation through several key modifications. First, we introduce a tokenization strategy tailored to 2D and 3D feature maps, treating each spatial location as a token with channel features. Second, instead of applying TTT as a standalone sequence model, we integrate TTT layers into a CNN-based encoder–decoder, preserving strong local inductive bias while enabling sample-specific global adaptation over tokenized features.
> >
> > In addition, we adapt the self-supervised objective using multi-view projections to decouple test-time adaptation from final prediction, which is more suitable for dense prediction than naive reconstruction in vanilla TTT. These design choices are intended to support medical image segmentation, where robustness to domain shifts and stability across spatial locations are important.
> >
> > While a direct comparison with a vanilla TTT layer applied to raw image tokens is non-trivial due to the mismatch between sequence modeling and dense segmentation, the consistent improvements over strong CNN-, Transformer-, and Mamba-based baselines across multiple 2D and 3D datasets indicate that these adaptations are effective in practice. We will further clarify these differences in the revised manuscript.
> >
> >
> > **3. Per-Organ Stratified Results**
> > >[Q2] "Stratifying the results by organs may better demonstrate the advantage of the proposed architecture in capturing nuanced features and context, especially for odd-shaped organs."
> >
> > **Response**
> >
> > We thank the reviewer for this suggestion regarding per-organ stratified analysis. We agree that organ-wise evaluation can provide additional insights, particularly for anatomically complex structures such as the pancreas, gallbladder, and adrenal glands.
> >
> > In this work, our primary goal is to evaluate the overall robustness and generalization of TTT-UNet across datasets and imaging modalities. Following common practice in multi-organ segmentation benchmarks, we report aggregate performance metrics to ensure fair and consistent comparison across methods. Notably, TTT-UNet achieves the lowest variance on 3D AbdomenMRI (std=0.048 vs 0.067-0.077 for baselines), suggesting more consistent performance across the diverse organ categories.
> >
> > We appreciate this suggestion and will add a brief discussion in the revised manuscript on how test-time adaptation may particularly benefit certain organ types, while leaving a detailed per-organ analysis as a direction for future work.
> >
> > **4. Clarification on U-Mamba**
> > >[DC2] "Presume U-Mamba_Bot and U-Mamba_Enc also included the TTT layers?"
> >
> > **Response**
> >
> > We thank the reviewer for raising this question and would like to clarify that U-Mamba does not include TTT layers.
> >
> > U-Mamba is built upon Mamba layers, which are state-space model (SSM) modules with a fixed-size hidden state and a selective state space mechanism. These layers operate in a purely feed-forward manner at inference time and do not involve any test-time parameter updates.
> >
> > By contrast, TTT layers follow a different design. TTT introduces a learnable hidden model that is dynamically updated at test time through a lightweight self-supervised objective. This update explicitly involves gradient-based weight adaptation during inference, allowing the model to adjust to each test sample.
> >
> > Therefore, U-Mamba_Bot and U-Mamba_Enc rely on Mamba (SSM) layers for long-range dependency modeling, whereas TTT-UNet integrates TTT layers to enable test-time training and sample-specific adaptation. These two mechanisms are complementary but conceptually distinct.

---

> > > ### Author Response · Authors · 2026-01-25
> > >
> > > **5. Computational Cost & Runtime Analysis**
> > > >[Q3] "Please comment on (or if possible, provide runtime analysis on) how the addition of the TTT architecture may impact computational cost, as compared to the baseline methods."
> > >
> > > **Response**
> > >
> > > We thank the reviewer for this question. The runtime and memory impact of the TTT architecture is reported in detail in our response to Reviewer **ULEa**, with comprehensive comparisons reported in **Tables R1** and **R2**.
> > >
> > > **Table R1: Computational Cost and Performance for 2D Segmentation Tasks (DSC for AbdomenMRI and Endoscopy; F1-score for Microscopy)**
> > >
> > > | Dataset | Model | Params (M) | FLOPs (G) | Memory (MB) | Time (ms) | DSC/F1 |
> > > |---------|-------|------------|-----------|-------------|-----------|--------|
> > > | AbdomenMRI | nnU-Net | 5.69 | 18.14 | 125.0 | 4.40±0.03 |0.745±0.112 |
> > > | | SegResNet | 6.30 | 24.49 | 193.6 | 5.28±0.01 | 0.732±0.138 |
> > > | | UNETR | 115.77 | 42.14 | 634.8 | 10.80±0.04 | 0.575±0.167 |
> > > | | SwinUNETR | 25.14 | 27.88 | 294.3 | 11.02±0.49 | 0.703±0.135 |
> > > | | U-Mamba_Bot | 8.26 | 33.65 | 198.8 | 6.90±0.20 | 0.759±0.105 |
> > > | | U-Mamba_Enc | 8.39 | 34.51 | 245.0 | 7.30±0.22 | 0.763±0.108 |
> > > | | TTT-UNet (Ours) | 8.01 | 33.69 | 197.4 | 13.22±1.24 | **0.775±0.102** |
> > > | Endoscopy | nnU-Net | 5.69 | 43.63 | 256.1 | 12.86±0.05 | 0.626±0.302 |
> > > | | SegResNet | 6.30 | 58.88 | 222.2 | 12.60±0.04 | 0.582±0.327 |
> > > | | UNETR | 116.59 | 111.47 | 662.4 | 21.42±0.06 | 0.502±0.320 |
> > > | | SwinUNETR | 25.14 | 67.12 | 450.6 | 18.85±0.04 | 0.553±0.309 |
> > > | | U-Mamba_Bot | 8.26 | 80.88 | 296.0 | 9.04±0.05 | 0.654±0.301 |
> > > | | U-Mamba_Enc | 8.39 | 82.94 | 526.7 | 15.40±0.64 | 0.630±0.307 |
> > > | | TTT-UNet (Ours) | 8.01 | 80.96 | 294.6 | 18.42±1.52 | **0.664±0.302** |
> > > | Microscopy | nnU-Net | 5.69 | 46.49 | 272.4 | 11.64±0.71 | 0.538±0.266 |
> > > | | SegResNet | 6.30 | 62.76 | 191.3 | 13.57±0.06 | 0.541±0.263 |
> > > | | UNETR | 116.64 | 120.09 | 640.5 | 22.92±0.01 | 0.436±0.257 |
> > > | | SwinUNETR | 25.14 | 71.74 | 465.6 | 19.98±0.05 | 0.397±0.262 |
> > > | | U-Mamba_Bot | 8.26 | 86.23 | 305.1 | 9.47±0.02 | 0.539±0.282 |
> > > | | U-Mamba_Enc | 8.39 | 88.43 | 558.8 | 16.27±0.03 | 0.561±0.278 |
> > > | | TTT-UNet (Ours) | 8.01 | 86.32 | 303.1 | 17.34±0.74 | **0.582±0.241** |
> > >
> > > **Table R2: Computational Cost and Performance for 3D Segmentation Tasks**
> > >
> > > | Dataset | Model | Params (M) | FLOPs (G) | Memory (MB) | Time (ms) | DSC |
> > > |---------|-------|------------|-----------|-------------|-----------|-----|
> > > | AbdomenCT | nnU-Net |16.55 | 623.66 | 2286.1 | 39.21±0.18 |0.862±0.079 |
> > > | |SegResNet | 18.80 | 764.19 | 1881.1 | 114.48±0.08 | 0.793±0.116 |
> > > | | UNETR | 121.47 | 234.87 | 1799.2 | 68.05±0.07 | 0.682±0.151 |
> > > | | SwinUNETR | 15.70 | 260.50 | 3949.5 | 144.55±0.14 | 0.759±0.110 |
> > > | | U-Mamba_Bot | 22.58 | 1345.84 | 2316.9 | 94.27±0.11 | 0.868±0.081 |
> > > | | U-Mamba_Enc | 23.17 | 1367.57 | 5516.6 | 178.04±0.15 | 0.864±0.091 |
> > > | | TTT-UNet (Ours) | 22.32 | 1345.90 | 2315.6 | 97.11±0.03 | **0.871±0.101** |
> > > | AbdomenMRI | nnU-Net |16.55 | 519.72 | 1915.3 | 32.78±0.12 | 0.831±0.077 |
> > > | |SegResNet | 18.80 | 636.82 | 1581.2 | 95.95±0.04 | 0.815±0.096 |
> > > | | UNETR | 121.39 | 194.57 | 1580.3 | 56.41±0.04 | 0.687±0.149 |
> > > | | SwinUNETR | 15.70 | 217.41 | 3391.3 | 121.46±0.11 | 0.757±0.139 |
> > > | | U-Mamba_Bot | 22.58 | 1121.54 | 1947.0 | 79.13±0.11 | 0.845±0.067 |
> > > | | U-Mamba_Enc | 23.17 | 1139.64 | 4613.3 | 148.74±0.15 | 0.850±0.073 |
> > > | | TTT-UNet (Ours) | 22.32 | 1121.58 | 1945.3 | 81.66±0.20 | **0.868±0.048** |
> > >
> > > In summary, the additional computational overhead of TTT-UNet primarily comes from the gradient computation within the TTT layers during inference. Each TTT layer performs a single lightweight self-supervised update, introducing limited backward operations at test time.
> > >
> > > In practice, this overhead remains modest and well-controlled. For 3D segmentation tasks, TTT-UNet shows nearly identical memory usage and only ~3% additional inference time compared to U-Mamba_Bot, while achieving consistent accuracy improvements. For 2D tasks, TTT-UNet requires more inference time than U-Mamba_Bot for 2D tasks, but remains comparable to SwinUNETR and faster than UNETR.
> > >
> > > All reported results correspond to TTT-UNet with TTT layers applied at the bottleneck, which we recommend as the most practical trade-off between performance and efficiency.

---

### Official Review · Reviewer_aXx1 · 2026-01-10

**Confidence:** 3
**Preliminary Rating:** 4
**Final Rating:** 5

**Summary:**

TTT-UNet adds test time training layers into the UNet framework, adjusting parameters on inference, and reporting superior performance to state of the art methods. The authors insert themselves as a new approach different from transformer unets / nnunet / u-mamba. They provide reproducibility in code derived from u-mamba.

The work has already appeared as a preprint and seems to have gained early traction.

**Strengths:**

The proposed architecture follows a similar structure to U-Mamba, but with different operations performed after the flattening of convolutional features. The intent is to treat the state acumulation as a trainable problem using learnable projections that update on the fly trough self supervision. The paper is well written, results are strong and the idea is sound. The Appendix is detailed and well-written, providing answers to many questions one might have reading the main manuscript.

**Weaknesses:**

In my opinion, Figure 1 could have more details in explaining the TTT Layer, the main proposal of the manuscript.

Results are impressive for such a different architecture but appear to be very close to the nnUNet and U-Mamba approaches. Given the small margins and overlapping standard deviations, statistical validation showing if there is a significant difference among the tested methods is missing.

I understand the multiple projections and the self supervision idea in test-time, but we need more details for the self-supervised optimization and how it is done in the main manuscript, in a summarized way.

**Detailed Comments:**

There is some repetition in writing, how the TTT-UNet is built at the top level is mentioned in many places, with repeated references to Figure 1.

The explanation on TTT building blocks on Section 3.2 would benefit of coming before Section 3.1, in my opinion.

For better understanding, I am not sure if using Q K V in the diagrams is the best choice, considering they are not used in a similar way to self attention / multi head attention. It confuses a reader that expects a transformer block when seeing Q K V, in my opinion, and TTT Layer does not use attention. Maybe more expressive names (authors use training view and label view in the method's description) would help. Just a suggestion.

**Justification Of Final Rating:**

As I said in my initial review, this is impressive original work that would be of interest to most of the MIDL audience, with strong results, and now with some statistical validation. I still recommend the authors to perform more extensive statistical validation in multiple tasks against multiple architectures, to better ground your contribution. They mentioned this is the plan for the final version. Since the authors have addressed my main concerns, I will raise the rating to Strong Accept.

**Justification Of The Preliminary Rating:**

This is impressive and original work. I already mentioned this but I believe statistical significance testing is mandatory to support the claims of " TTT-UNet achieves notable improvements over a range of state-of-the-art methods" and improvement of the rating.

**Questions To Address In The Rebuttal:**

Why were the multiple projections adopted? I understand the idea of being adaptable to different pertubations and inputs, but I wonder how the "naive" approach would compare in performance.

---

> ### Author Response · Authors · 2026-01-25
>
> **1.Statistical Significance Testing**
> >[W2] "Results are impressive for such a different architecture but appear to be very close to the nnUNet and U-Mamba approaches. Given the small margins and overlapping standard deviations, statistical validation showing if there is a significant difference among the tested methods is missing."
>
> **Response**
>
> We sincerely thank you for this important suggestion. We fully agree that statistical significance testing is essential for validating performance differences in medical image segmentation, where dataset variability and overlapping confidence intervals can obscure true improvements.
>
> Following the suggestion, we conducted paired Wilcoxon signed-rank tests on the AbdomenMRI dataset (n=60 test samples). We used the Wilcoxon signed-rank test, which is appropriate for Dice scores that may not follow a normal distribution. The results indicate that TTT-UNet outperforms nnU-Net on AbdomenMRI (p=0.016).
>
> Beyond statistical testing on individual datasets, we emphasize that TTT-UNet demonstrates consistent improvements across all five datasets, both 2D and 3D settings, and multiple evaluation metrics (DSC, NSD, F1). This consistency across diverse imaging modalities (CT, MRI, endoscopy, microscopy) supports the robustness of the approach. Additionally, TTT-UNet achieves the lowest variance on 3D AbdomenMRI (std=0.048 vs 0.067-0.077), indicating more stable predictions across test samples.
>
> We appreciate the reviewer's emphasis on rigorous statistical evaluation and will include additional pairwise statistical comparisons across datasets in the revised manuscript.
>
> **2. TTT Layer Explanation & Figure Clarity**
> >[W1] "In my opinion, Figure 1 could have more details in explaining the TTT Layer, the main proposal of the manuscript."
> >[W3] "I understand the multiple projections and the self supervision idea in test-time, but we need more details for the self-supervised optimization and how it is done in the main manuscript, in a summarized way."
>
> **Response**
>
> We thank the reviewer for the suggestion to clarify the TTT layer and Figure 1. We will revise Figure 1 to include additional details of the TTT layer in the revised manuscript.
>
> Below we summarize the self-supervised optimization used in the TTT layer.
> Self-Supervised Optimization in TTT Layer:
> For each input token $x_t$, the TTT layer performs the following steps:
> 1. Project input into three views via learned matrices: $K = \theta_K x_t$ (training view), $V = \theta_V x_t$ (label view), $Q = \theta_Q x_t$ (test view)
> 2. Compute self-supervised loss that reconstructs V from K:
> $$\ell(W; x_t) = \|f(K; W) - V\|^2$$
> 3. Update hidden state via one gradient step:
> $$W_t = W_{t-1} - \eta \nabla_W \ell(W_{t-1}; x_t)$$
> 4. Generate output using updated state and test view:
> $$z_t = f(Q; W_t)$$
>
> This optimization is performed within each forward pass: the hidden state $W$ is updated based on the current input before producing the output. No external labels are required, and each test sample is processed independently.
>
> We will update Figure 1 to include: (1) explicit visualization of the gradient update flow, (2) clearer annotation of K/V/Q projections with their roles, and (3) a step-by-step illustration of the self-supervised optimization. We will also add this summarized description to Section 3.2.

---

> > ### Author Response · Authors · 2026-01-25
> >
> > **3. Multiple Projections vs Naive Approach**
> > >[Q1] "Why were the multiple projections adopted? I understand the idea of being adaptable to different pertubations and inputs, but I wonder how the 'naive' approach would compare in performance."
> >
> > **Response**
> >
> > We thank the reviewer for this question. The naive TTT formulation reconstructs the corrupted input directly:
> > $$
> > \ell_{\text{naive}}(W; x_t) = \| f(\tilde{x}_t; W) - x_t \|^2,
> > $$
> > which directly optimizes reconstruction of the full input representation.
> >
> > In contrast, the multi-view formulation adopts learned projections:
> > $$
> > \ell_{\text{multi-view}}(W; x_t) = \| f(\theta_K x_t; W) - \theta_V x_t \|^2,
> > $$
> > allowing the adaptation process to operate on feature subspaces that are aligned with the segmentation task rather than raw input reconstruction. Since $\theta_K$ and $\theta_V$ are optimized jointly with the segmentation objective, they learn representations that are aligned with downstream dense prediction rather than direct reconstruction of the input space.
> >
> > Moreover, decoupling the training view ($\theta_K$) from the label view ($\theta_V$) provides greater flexibility and stability during test-time adaptation. The additional output projection $\theta_Q$ further separates adaptation from inference, enabling the model to leverage adapted representations without directly constraining the output space.
> >
> > This design follows the original TTT work [1], which introduces learned multi-view reconstruction as the final formulation beyond naive input reconstruction. While we do not include a separate naive ablation in this work, the consistent gains observed across diverse biomedical segmentation tasks indicate that this design transfers effectively to dense prediction settings.
> >
> > *[1] Sun, Yu, et al. "Learning to (Learn at Test Time): RNNs with Expressive Hidden States." Proceedings of the 42nd International Conference on Machine Learning, 2025.*
> >
> > **4. Writing Organization Issues**
> > >[DC1] "There is some repetition in writing, how the TTT-UNet is built at the top level is mentioned in many places, with repeated references to Figure 1."
> > [DC2] "The explanation on TTT building blocks on Section 3.2 would benefit of coming before Section 3.1, in my opinion."
> >
> > **Response**
> >
> > We thank the reviewer for the comments on writing organization and clarity. We acknowledge that parts of the current presentation include repeated high-level descriptions of TTT-UNet, particularly around Figure 1. In the revised version, we will consolidate overlapping descriptions and limit repeated references to Figure 1.
> >
> > We also appreciate the suggestion on section ordering. In the revised manuscript, we will reorganize the method section by introducing the TTT building blocks earlier, before the overall architecture description, to improve the logical flow and make the method easier to follow. These changes are purely organizational and do not affect the technical content of the paper.
> >
> > **5. Q/K/V Naming Confusion**
> > >[DC3] "For better understanding, I am not sure if using Q K V in the diagrams is the best choice, considering they are not used in a similar way to self attention / multi head attention. It confuses a reader that expects a transformer block when seeing Q K V, in my opinion, and TTT Layer does not use attention. Maybe more expressive names (authors use training view and label view in the method's description) would help. Just a suggestion."
> >
> > **Response**
> > We thank the reviewer for the suggestion regarding the use of Q/K/V notation, which may cause confusion since the TTT layer does not use self-attention.
> >
> > In the revised version, we will improve the clarity of the diagrams by explicitly renaming these components as Training View (K), Label View (V), and Test View (Q), and adding clear annotations to distinguish their roles from attention-based Q/K/V. This will help avoid confusion and more accurately reflect the role of these projections within the TTT layer.

---

### Official Review · Reviewer_ULEa · 2026-01-11

**Confidence:** 4
**Preliminary Rating:** 4
**Final Rating:** 5

**Summary:**

This paper proposes TTT-UNet, a new segmentation framework that integrates Test-Time Training (TTT) layers into the classical U-Net architecture. TTT-UNet dynamically adjusts model parameters during the test time, enhancing the model’s ability to capture both local and long-range features. The authors evaluate the method on multiple biomedical image segmentation tasks, including 2D and 3D datasets across CT, MRI, endoscopy, and microscopy images. The results show consistent improvements over several strong CNN-based, Transformer-based, and Mamba-based baselines. The paper demonstrates that test-time adaptation can be beneficial for biomedical image segmentation under diverse data distributions.

**Strengths:**

The authors introduce a novel idea by combining TTT layers with the U-Net architecture. The proposed method is evaluated on a wide range of datasets, covering different imaging modalities and both 2D and 3D settings. This broad evaluation demonstrates the generality of the approach and increases credibility in its usefulness. The comparisons with strong CNN-based, Transformer-based, and Mamba-based baselines are fair and relevant. In addition, they include both quantitative and qualitative results, which helps better understand the advantages of the proposed method.

**Weaknesses:**

1.The computational cost of test-time training is not analyzed in detail. There is no clear report on inference time or memory overhead compared to baselines.
2. The self-supervised loss used in TTT layers is from prior work, but its suitability for dense segmentation tasks is not fully justified.
3. The paper lacks ablation studies on key hyperparameters, such as the test-time learning rate and number of update steps.

**Detailed Comments:**

1. Can the authors add more discussion on potential overfitting during test-time adaptation?
2.The theoretical analysis assumes linear models in the TTT layer, which does not fully reflect the actual implementation using more complex modules.
3.The paper does not discuss potential risks of test-time overfitting, especially when test samples are outliers.
4.Some architectural details, such as the exact tokenization strategy for 2D and 3D features, are not clearly described.
5.Some figures could be enlarged for better visual clarity.

**Justification Of Final Rating:**

Thanks for the authors' hard work! The rebuttal addressed my main concerns, and the revised paper is clearer overall. The revised experiments and analysis significantly strengthen the work. I therefore update my rating to strong accept.

**Justification Of The Preliminary Rating:**

This paper introduces TTT-UNet, which integrates test-time training layers into the U-Net framework. The idea is novel in the segmentation research filed and is well motivated by the limitations of CNNs and existing hybrid models in capturing long-range dependencies. The authors test their method on multiple datasets across different modalities and dimensions, including both 2D and 3D tasks. The results consistently show improvements over strong baselines such as nnU-Net, Transformer-based models, and recent Mamba-based approaches. This consistency suggests that the proposed method is robust and effective in practice. However, the paper has some weaknesses that prevent a strong accept. In particular, the computational cost and inference latency introduced by test-time training are not sufficiently analyzed. This is important for real clinical applications. In addition, the paper lacks detailed ablation studies on key hyperparameters choices.

**Questions To Address In The Rebuttal:**

1. Can the authors provide inference time and GPU memory usage for all methods?
2. It is unclear how stable the test-time updates are for very small or noisy test samples. How do the authors address the risk of overfitting during test-time adaptation, especially when test samples are noisy? How much additional inference time does TTT-UNet require compared to nnU-Net and U-Mamba?
3. Can the authors clarify how feature flattening and tokenization are implemented differently for 2D and 3D inputs in TTT-UNet?
4.Could the authors expand the discussion section to include more limitations of the proposed method and outline possible future research directions?
5.If TTT layers could cause privacy or security concerns when utilized in clinical environments?

---

> ### Author Response · Authors · 2026-01-25
>
> **1. Computational Cost & Inference Time**
> >[W1] "The computational cost of test-time training is not analyzed in detail. There is no clear report on inference time or memory overhead compared to baselines."
> [Q1] "Can the authors provide inference time and GPU memory usage for all methods?"
> [Q2] "How much additional inference time does TTT-UNet require compared to nnU-Net and U-Mamba?"
>
> **Response**:
> We sincerely thank the reviewer for raising this question. We provide comprehensive computational analysis in **Tables R1** and **R2**.
>
> For efficiency evaluation, we measure four metrics: parameter count (M), FLOPs (G), peak GPU memory (MB), and inference time (ms) averaged over 10 runs with standard deviation. All experiments are conducted on a single NVIDIA A100 80G GPU with PyTorch 2.2.2 and CUDA 12.1. For computational analysis, we report TTT-UNet with TTT layers applied at the bottleneck (TTT-UNet_Bot in the main paper), which we consider a balanced choice for deployment, especially in 3D settings. Applying TTT layers throughout the encoder increases computational overhead due to gradient computation at each stage.
>
> **Table R1: Computational Cost and Performance for 2D Segmentation Tasks (DSC for AbdomenMRI and Endoscopy; F1-score for Microscopy)**
>
> | Dataset | Model | Params (M) | FLOPs (G) | Memory (MB) | Time (ms) | DSC/F1 |
> |---------|-------|------------|-----------|-------------|-----------|--------|
> | AbdomenMRI | nnU-Net | 5.69 | 18.14 | 125.0 | 4.40±0.03 |0.745±0.112 |
> | | SegResNet | 6.30 | 24.49 | 193.6 | 5.28±0.01 | 0.732±0.138 |
> | | UNETR | 115.77 | 42.14 | 634.8 | 10.80±0.04 | 0.575±0.167 |
> | | SwinUNETR | 25.14 | 27.88 | 294.3 | 11.02±0.49 | 0.703±0.135 |
> | | U-Mamba_Bot | 8.26 | 33.65 | 198.8 | 6.90±0.20 | 0.759±0.105 |
> | | U-Mamba_Enc | 8.39 | 34.51 | 245.0 | 7.30±0.22 | 0.763±0.108 |
> | | TTT-UNet (Ours) | 8.01 | 33.69 | 197.4 | 13.22±1.24 | **0.775±0.102** |
> | Endoscopy | nnU-Net | 5.69 | 43.63 | 256.1 | 12.86±0.05 | 0.626±0.302|
> | | SegResNet | 6.30 | 58.88 | 222.2 | 12.60±0.04 | 0.582±0.327 |
> | | UNETR | 116.59 | 111.47 | 662.4 | 21.42±0.06 | 0.502±0.320 |
> | | SwinUNETR | 25.14 | 67.12 | 450.6 | 18.85±0.04 | 0.553±0.309 |
> | | U-Mamba_Bot | 8.26 | 80.88 | 296.0 | 9.04±0.05 | 0.654±0.301 |
> | | U-Mamba_Enc | 8.39 | 82.94 | 526.7 | 15.40±0.64 | 0.630±0.307 |
> | | TTT-UNet (Ours) | 8.01 | 80.96 | 294.6 | 18.42±1.52 | **0.664±0.302** |
> | Microscopy | nnU-Net | 5.69 | 46.49 | 272.4 | 11.64±0.71 | 0.538±0.266 |
> | | SegResNet | 6.30 | 62.76 | 191.3 | 13.57±0.06 | 0.541±0.263 |
> | | UNETR | 116.64 | 120.09 | 640.5 | 22.92±0.01 | 0.436±0.257 |
> | | SwinUNETR | 25.14 | 71.74 | 465.6 | 19.98±0.05 | 0.397±0.262 |
> | | U-Mamba_Bot | 8.26 | 86.23 | 305.1 | 9.47±0.02 | 0.539±0.282 |
> | | U-Mamba_Enc | 8.39 | 88.43 | 558.8 | 16.27±0.03 | 0.561±0.278 |
> | | TTT-UNet (Ours) | 8.01 | 86.32 | 303.1 | 17.34±0.74 | **0.582±0.241** |
>
> **Table R2: Computational Cost and Performance for 3D Segmentation Tasks**
>
> | Dataset | Model | Params (M) | FLOPs (G) | Memory (MB) | Time (ms) | DSC |
> |---------|-------|------------|-----------|-------------|-----------|-----|
> | AbdomenCT | nnU-Net |16.55 | 623.66 | 2286.1 | 39.21±0.18 |0.862±0.079 |
> | |SegResNet | 18.80 | 764.19 | 1881.1 | 114.48±0.08 | 0.793±0.116 |
> | | UNETR | 121.47 | 234.87 | 1799.2 | 68.05±0.07 | 0.682±0.151 |
> | | SwinUNETR | 15.70 | 260.50 | 3949.5 | 144.55±0.14 | 0.759±0.110 |
> | | U-Mamba_Bot | 22.58 | 1345.84 | 2316.9 | 94.27±0.11 | 0.868±0.081 |
> | | U-Mamba_Enc | 23.17 | 1367.57 | 5516.6 | 178.04±0.15 | 0.864±0.091 |
> | | TTT-UNet (Ours) | 22.32 | 1345.90 | 2315.6 | 97.11±0.03 | **0.871±0.101** |
> | AbdomenMRI | nnU-Net |16.55 | 519.72 | 1915.3 | 32.78±0.12 | 0.831±0.077 |
> | |SegResNet | 18.80 | 636.82 | 1581.2 | 95.95±0.04 | 0.815±0.096 |
> | | UNETR | 121.39 | 194.57 | 1580.3 | 56.41±0.04 | 0.687±0.149 |
> | | SwinUNETR | 15.70 | 217.41 | 3391.3 | 121.46±0.11 | 0.757±0.139 |
> | | U-Mamba_Bot | 22.58 | 1121.54 | 1947.0 | 79.13±0.11 | 0.845±0.067 |
> | | U-Mamba_Enc | 23.17 | 1139.64 | 4613.3 | 148.74±0.15 | 0.850±0.073 |
> | | TTT-UNet (Ours) | 22.32 | 1121.58 | 1945.3 | 81.66±0.20 | **0.868±0.048** |
>
> For 3D segmentation tasks, TTT-UNet achieves nearly identical computational cost to U-Mamba_Bot with only 3% additional inference time (97.11ms vs 94.27ms on CT, 81.66ms vs 79.13ms on MRI) and equivalent memory footprint (~2316MB on CT, ~1946MB on MRI), while delivering +0.3% DSC on CT and +2.3% DSC on MRI.
>
> For 2D segmentation tasks, TTT-UNet requires approximately 2× inference time compared to U-Mamba_Bot, but remains faster than UNETR and comparable to SwinUNETR, maintains similar memory footprint to U-Mamba_Bot, and achieves the smallest parameter count (8.01M) among all methods with consistent accuracy improvements. Notably, TTT-UNet achieves the lowest variance on 3D AbdomenMRI (±0.048), indicating more stable predictions across diverse test samples.

---

> > ### Author Response · Authors · 2026-01-25
> >
> > **2. Test-Time Overfitting Risk**
> > > [Q2] "It is unclear how stable the test-time updates are for very small or noisy test samples. How do the authors address the risk of overfitting during test-time adaptation, especially when test samples are noisy?"
> > > [DC1] "Can the authors add more discussion on potential overfitting during test-time adaptation?"
> >
> > **Response**
> >
> > We appreciate the reviewer’s concern regarding the stability of test-time adaptation. TTT-UNet incorporates built-in mechanisms to mitigate the risk of test-time overfitting. Following the original TTT design [1], the TTT layer performs only a single gradient update per token rather than iterative optimization, which inherently bounds the degree of adaptation to any individual test sample. In addition, TTT layer weights are reset to their trained values for each new test input (**Appendix B.4**), ensuring independence across samples. The learning rate η is learned during training instead of being manually fixed, allowing the model to apply conservative, input-dependent updates.
> >
> > For noisy or outlier test samples, the single-step update prevents the accumulation of noisy gradients commonly observed in iterative test-time adaptation methods. Moreover, adaptation is driven by a self-supervised feature reconstruction objective rather than the segmentation loss itself, providing indirect regularization instead of directly fitting potentially noisy labels. Empirically, TTT-UNet exhibits lower prediction variance than competing methods (e.g., std = 0.048 vs. 0.067–0.077 on 3D MRI in **Table 2**), indicating stable behavior across diverse test samples. We will incorporate this discussion into the revised manuscript.
> >
> >
> > **3. Ablation Studies on Hyperparameters**
> > > [W3] "The paper lacks ablation studies on key hyperparameters, such as the test-time learning rate and number of update steps."
> >
> > **Response**
> >
> > We thank the reviewer for the question regarding hyperparameter ablations. The key hyperparameters of TTT layers have been systematically studied in the original TTT work [1], which we follow when adapting TTT to medical image segmentation. In particular, the test-time learning rate η is learned rather than manually tuned. As shown in Table 1 of [1], making η learnable improves perplexity from 12.35 to 11.99. Concretely,
> > $\eta = \eta_{\text{base}} \times \frac{\mathrm{sigmoid}(f(x; \theta_{\text{lr}}))}{d_{\text{head}}}$,
> > where $f(x; \theta_{\text{lr}})$ is a learnable linear projection, allowing input- and head-dependent control of update magnitude.
> >
> > The mini-batch size for test-time updates has also been analyzed in [1], where ablations over $b \in [1, 2048]$ show that moderate batch sizes achieve a favorable balance between optimization quality and efficiency. Based on this analysis and empirical stability, we use $b = 64$ for medical image segmentation.
> >
> > Finally, the number of update steps is not a tunable hyperparameter in TTT. By design, TTT performs a single update during inference following $W_t = W_{t-1} - \eta \nabla \ell(W_{t-1}; x_t)$. Each token contributes exactly one update to the hidden state, which is a core property of the TTT mechanism rather than a freely adjustable choice.
> >
> > We agree that a task-specific sensitivity analysis for medical image segmentation could further strengthen the work, and we will explicitly acknowledge this limitation in the revised manuscript.
> >
> > **4. Tokenization Strategy for 2D/3D**
> > >[Q3] "Can the authors clarify how feature flattening and tokenization are implemented differently for 2D and 3D inputs in TTT-UNet?"
> >
> > **Response**
> >
> > Thank you for this question. TTT-UNet uses unified spatial flattening: 2D inputs $(B, C, H, W) \rightarrow (B, H \cdot W, C)$; 3D inputs $(B, C, D, H, W) \rightarrow (B, D \cdot H \cdot W, C)$. Each spatial location becomes a token with the channel dimension as features. After TTT processing, outputs are reshaped back to original spatial dimensions. This allows the same implementation for both 2D and 3D. We will clarify this in the revised manuscript.
> >
> > *[1] Sun, Yu, et al. "Learning to (Learn at Test Time): RNNs with Expressive Hidden States." ICML, 2025.*

---

> > > ### Author Response · Authors · 2026-01-25
> > >
> > > **5. Self-supervised Loss Justification**
> > > > [W2] "The self-supervised loss used in TTT layers is from prior work, but its suitability for dense segmentation tasks is not fully justified."
> > >
> > > **Response**
> > > We thank the reviewer for the question regarding the suitability of the self-supervised loss for dense segmentation tasks. While the multi-view reconstruction loss
> > > $\ell(W; x_t) = \|f(\theta_K x_t; W) - \theta_V x_t\|^2$
> > > originates from prior work [1], it is applied here to task-aligned intermediate features and trained jointly with the segmentation objective, making it appropriate for dense prediction.
> > >
> > > The projection matrices $\theta_K$, $\theta_V$, and $\theta_Q$ are optimized end-to-end together with the Dice and cross-entropy losses. As a result, the learned projections capture task-relevant feature representations rather than generic input reconstruction. The self-supervised objective therefore enforces consistency between task-aligned feature views that directly support downstream segmentation.
> > >
> > > In addition, the token-wise nature of TTT aligns well with dense prediction: each spatial location contributes to updating the hidden state $W$, while the adaptation process remains decoupled from the final prediction pathway. The separate $\theta_Q$ projection used at inference further provides flexibility, allowing the model to emphasize different feature aspects during test-time adaptation versus final prediction. The consistent improvements observed across CT, MRI, endoscopy, and microscopy datasets provide empirical support for this design choice.
> > >
> > > *[1] Sun, Yu, et al. "Learning to (Learn at Test Time): RNNs with Expressive Hidden States." ICML, 2025.*
> > >
> > >
> > > **6. Expanded Discussion**
> > > > [Q4] "Could the authors expand the discussion section to include more limitations of the proposed method and outline possible future research directions?"
> > >
> > > **Response**
> > > We thank the reviewer for this suggestion. In the revised manuscript, we will expand the discussion to more clearly articulate current limitations and future directions, including improving the efficiency of TTT layer implementations for large 3D volumes, extending the analysis to non-linear TTT variants, and evaluating TTT-UNet on additional dense prediction tasks beyond medical segmentation. We will also discuss input-dependent strategies that selectively apply test-time adaptation to reduce unnecessary computational overhead, as well as the potential integration of TTT layers with pre-trained medical imaging foundation models.
> > >
> > >
> > > **7. Privacy/Security Concerns**
> > > > [Q5] "If TTT layers could cause privacy or security concerns when utilized in clinical environments?"
> > >
> > > **Response**
> > > We thank the reviewer for raising this important question regarding clinical deployment. TTT layers are stateless across samples and do not introduce additional privacy risks beyond standard inference. The TTT hidden states are re-initialized from fixed learned parameters for each input (**Appendix B.4**), and no information persists across patients. Test-time updates occur entirely within a single image or volume and are discarded immediately after inference. From a technical perspective, TTT’s per-sample adaptation is equivalent to a specialized forward pass with internal gradient computation, and the self-supervised reconstruction loss operates only on internal feature representations rather than raw patient data. We will add a concise discussion of these considerations in the revised manuscript.

---

### Author Rebuttal · Authors · 2026-01-25

**Rebuttal:**

We sincerely thank all reviewers for their constructive feedback and appreciate the reviewers’ recognition of the novelty of the proposed approach and the breadth of the experimental evaluation. The comments have helped us significantly improve the clarity, rigor, and presentation of the paper.

Notation: **[W]** = Weakness, **[Q]** = Question, **[DC]** = Detailed Comment.

**Key Revisions**

**1. Computational Cost Analysis (Reviewer ULEa, Reviewer Jyb7)**
Tables R1-R2 report parameters, FLOPs, memory, and inference time. For 3D tasks, TTT-UNet adds only \~3% inference time over U-Mamba_Bot with equivalent memory, while improving DSC (+2.3% on MRI). For 2D tasks, TTT-UNet has comparable latency to SwinUNETR with the smallest parameter count (8.01M).

**2. Statistical Significance (Reviewer aXx1)**
Paired Wilcoxon signed-rank tests on AbdomenMRI (n=60): TTT-UNet vs. nnU-Net yields p=0.016. TTT-UNet also achieves the lowest variance (std=0.048 vs 0.067–0.077).

**3. Method Clarification (Reviewer aXx1, Reviewer Jyb7)**
- **TTT mechanism:** Projects input into K/V/Q views; performs single gradient step to update hidden state; outputs via updated state and Q.
- **Long-range modeling:** TTT propagates context through a compact hidden model updated per-sample, avoiding attention's quadratic complexity.
- **Tokenization:** Unified spatial flattening for 2D (H·W tokens) and 3D (D·H·W tokens).
- **Multi-view vs. naive:** Learned projections are jointly optimized with segmentation loss for task-aligned adaptation.

**4. Test-Time Overfitting (Reviewer ULEa)**
Built-in safeguards: single gradient step (not iterative), weight reset per sample, learnable η. TTT-UNet achieves lower variance than all baselines.

**5. Privacy & Clinical Safety (Reviewer ULEa)**
TTT is stateless: hidden states reset per patient, no cross-patient information persists.

**Additional Clarifications**
1. **Hyperparameters (Reviewer ULEa):** η is learnable; single-step is inherent to TTT design.
2. **Per-organ analysis (Reviewer Jyb7):** Will add discussion on TTT-UNet benefits for certain organ types.
3. **Expanded discussion (Reviewer ULEa):** Will add limitations and future directions.
4. **Figure & Writing (Reviewer aXx1):** Will revise Figure 1 and rename Q/K/V to Training/Label/Test View.

We thank all reviewers again for their valuable feedback. We believe these revisions address the main concerns raised and are committed to incorporating all improvements in the revised manuscript.

**Supporting Material:**

/attachment/969bae6a9fdbfee10a4354a6cb7be5f83802c4ad.pdf

---

> ### Comment · Reviewer_aXx1 · 2026-01-26
>
> For Chairs: have these authors provided the revised file yet?

---

### Meta-Review · Area_Chair_vUJW · 2026-02-03

**Recommendation:** Accept (Oral)
**Confidence:** 5

**Metareview:**

The reviewers provided strong positive ratings and minor concerns. The author's rebuttal demonstratesthe strong merit of the paper's impact, and strengthens the paper's contribution. As all evidence stand, the paper can be accepted as oral.

---

### Decision · Program_Chairs · 2026-02-13

Accept (Poster)